# Bures-Isotropy Alignment: Manifold Learning of Generalized Category Discovery

**Luyao Tang**[1*], **Kunze Huang**[2*], **Chaoqi Chen**[3], **Cheng Chen**[1†]

[1] The University of Hong Kong [2] Xiamen University [3] Shenzhen University

`lytang1999@gmail, cheng.chen@hku.hk`

## Abstract

Generalized Category Discovery (GCD) seeks to discover categories by clustering unlabeled samples that mix known and novel classes. While the prevailing recipe enforces compact clustering, this pursuit is largely blind to representation geometry: it over-compresses token manifolds, distorts eigen-structure, and yields brittle feature distributions that undermine discovery. We argue that GCD requires not more compression, but geometric restoration of an over-flattened feature space. Drawing inspiration from quantum information science, which similarly pursues representational completeness, we introduce **Bures-Isotropy Alignment** (BIA), which optimizes the mini-batch class-token Gram toward an isotropic prior by minimizing the Bures distance. Under a mild trace constraint, BIA admits a practical surrogate equivalent to maximizing the nuclear norm of stacked class tokens, thereby promoting isotropic, non-collapsed subspaces without altering architectures. The induced isotropy homogenizes the eigen-spectrum and raises the von Neumann entropy, improving both cluster separability and class-number estimation. BIA is plug-and-play, implemented in a few lines on unlabeled batches, and generally boosts strong GCD baselines on coarse- and fine-grained benchmarks, improving overall accuracy and reducing errors in the estimation of class-number. By restoring the geometry of token manifolds rather than compressing them blindly, BIA supplies compactness for known classes and cohesive emergence for novel ones, advancing robust open-world discovery. Code is available at github.com/lytang63/BIA.

## 1 Introduction

Open-world learning (Zhou et al., 2022; Wu et al., 2024) mandates models that can re-identify known classes while discovering novel ones from unlabeled data. Generalized category discovery (GCD) (Vaze et al., 2022b) formulates this demand explicitly, relaxing the assumptions of open-set recognition (OSR) (Geng et al., 2020) and novel-class discovery (NCD) (Han et al., 2019). Despite recent progress, the dominant paradigm in GCD still hinges on compact clustering as a universal target. This practice is geometry-agnostic: it compresses intra-class variability indiscriminately, often collapsing token manifolds into a few principal directions. The result is a degraded representation geometry with poor eigenstructure, skewed energy distribution, and fragile decision regions, that ultimately impedes category discovery and class-number estimation.

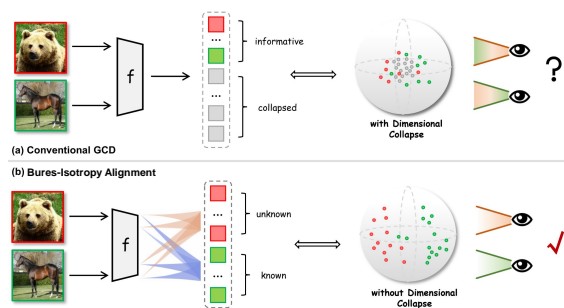

Figure 1: (a) GCD is constrained by dimensional collapse due to strong clustering, leading to mixed class features and limited representational capacity. (b) BIA enhances the token geometry capacity, improving representational completeness and unlocking the model's potential in the open world.

---

* Equal Contribution; † Corresponding Author.

We contend that the bottleneck is not insufficient compactness, but the loss of ***geometric quality*** in learned representations. When the class-token subspace is over-compressed, the feature distribution becomes anisotropic and low-rank, masking fine-grained semantics and amplifying open-world errors such as spurious merges. Thus, a central desideratum for GCD is to *restore a well-posed geometry and preserves intra-class completeness while maintaining inter-class separability*.

To this end, we propose **Bures-Isotropy Alignment** (**BIA**), a geometry-aware principle inspired by quantum information. As shown in Figure 1, green represents **known** classes and red represents **novel**[1] classes. The upper panel illustrates how conventional GCD methods, by enforcing compactness, cause dimensional collapse, leading to uninformative feature dimensions (grey area). BIA restores geometry, enhancing separability by encouraging isotropy. BIA aligns the empirical class-token covariance with an isotropic prior by minimizing the *Bures distance* (Jozsa, 1994), a canonical metric that instantiates the 2-Wasserstein geometry over covariance operators. Conceptually, BIA redistributes spectral energy across eigen-directions, prevents dimensional collapse, and yields uniform, full-rank token manifolds that capture richer intra-class semantics without sacrificing discrimination.

BIA comes with a practical and rigorous surrogate. Under mild row-norm or trace control, minimizing the Bures distance to identity is equivalent to *maximizing the nuclear norm of stacked class tokens*. This equivalence connects isotropy alignment with a simple few-line implementation that is architecture-agnostic and training-protocol compatible. It also bridges information geometry with GCD's capacity-oriented diagnostics: by homogenizing the spectrum, BIA increases the von Neumann entropy (Nielsen & Chuang, 2010) and the effective rank of the class-token autocorrelation, which we observe to correlate with more reliable class-number estimation and more stable clustering.

We integrate BIA into representative GCD frameworks (contrastive and prototype-based) without modifying backbones and loss schedules. BIA consistently improves All/Old/New accuracy across standard coarse- and fine-grained benchmarks, while reducing class-number estimation error. Ablations show that BIA stabilizes pseudo-label assignment in GCD, attenuates early collapse, and remains robust across backbones and batch sizes, at negligible computational overhead thanks to a Gram-matrix implementation (Borgwardt et al., 2006).

- We introduce BIA, casting GCD as isotropy alignment of class-token covariance via the Bures distance, thereby restoring representation geometry rather than compressing it blindly.

- We establish an equivalence between BIA and nuclear-norm maximization under a trace constraint, explaining BIA's effect on eigen-spectrum uniformity, von Neumann entropy, and effective rank, and linking information geometry to capacity-aware diagnostics in GCD.

- We provide a plug-and-play implementation that yields consistent gains on strong GCD baselines in both clustering accuracy and $K$-estimation, with minimal code and overhead.

## 2 RELATED WORKS

### 2.1 GENERALIZED CATEGORY DISCOVERY

Generalized category discovery (Vaze et al., 2022b; Zhao et al., 2023; Wen et al., 2023; Choi et al., 2024) jointly recognizes known classes and discovers unseen ones. The seminal framework (Vaze et al., 2022b) integrates semi-supervised k-means; SimGCD (Wen et al., 2023) introduces a parametric classifier with entropy regularization and self-distillation; CMS (Choi et al., 2024) enhances representations via mean-shift clustering; and (Zhao et al., 2023) adapts prototype counts at inference. ActiveGCD (Ma et al., 2024) further queries labels for selected unlabeled samples to improve discovery. Recent GCD works have advanced medical category discovery by reducing seen-class bias with neighbor-guided unbiased learning (Feng et al., 2025) and extending image-only discovery to multimodal image–text disease understanding (Feng et al., 2026a). Beyond static medical settings, frequency-guided GCD tackles domain-shifted discovery through spectral separation and perturbation (Feng & Ge, 2025), while PRISM further enables continual category discovery from evolving unlabeled streams (Feng et al., 2026b). Most GCD methods still prioritize compact clustering and thus overlook a central GCD d esideratum: restoring a well-posed geometry that preserves intra-class completeness while maintaining inter-class separability. We instead provide a concise, method-agnostic

---

[1]In the task setting of GCD, we do not differentiate between 'Old/New' and 'Known/Novel'.

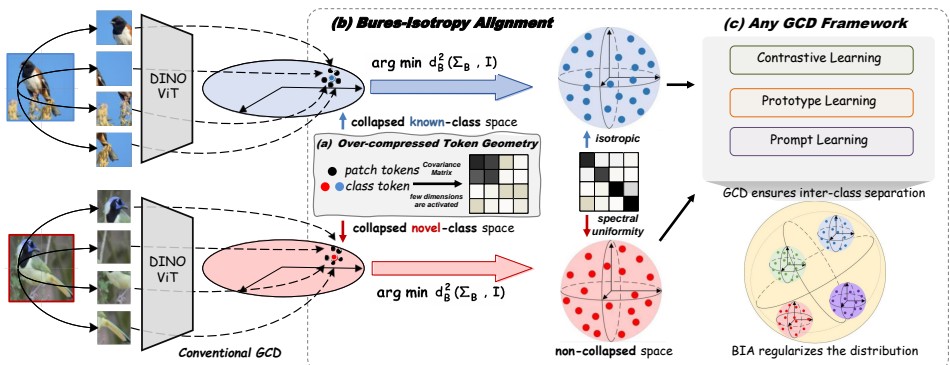

Figure 2: Overview of Bures-Isotropy Alignment. (a) BIA targets the geometry of the `[cls]` token, which summarizes sample information. (b) The core mechanism restores over-compressed, low-rank representations to a high-rank, isotropic state, preserving richer intra-class semantics. (c) The resulting embeddings are plug-and-play compatible with any GCD framework.

mechanism that enforces Bures-isotropy representations, thereby sharpening decision boundaries across GCD pipelines.

## 2.2 Representation Completeness

Representation completeness collapse (Grill et al., 2020; Caron et al., 2020; Shi et al., 2023; Jing et al., 2021) occurs when embeddings concentrate in a low-dimensional subspace, limiting diversity and expressiveness. DirectCLR (Jing et al., 2021) optimizes the representation space without a trainable projector to promote uniform dispersion. While Whitening (Tao et al., 2024) equalizes covariance contributions, the non-contrastive objective for collaborative filtering (Chen et al., 2024) emphasizes alignment and compactness without augmentation. Similarly, Bregman matrix divergence (Zhang et al., 2024) pulls covariance toward the identity, and random orthogonal projection modeling (Haghighat et al., 2023) broadens the search for characteristics. Rather than directly targeting collapse, we reformulate GCD as isotropy alignment of class-token covariance measured by the Bures distance, prioritizing the restoration of representation geometry over blind compression.

## 2.3 Bures Metric

The Bures metric (Šafránek, 2017) is based on the Bures distance and is originally defined to quantify the difference between quantum states. It is widely applied in the study of quantum information geometry (Cho & Jae, 2025), quantum physics (Alsing et al., 2023), and related areas. By interpreting density matrices as a generalized form of probability distributions, the Bures metric becomes an effective method for comparing probability distributions or positive definite matrices. For example, in (Ji et al., 2019), it is employed as an alternative to the Wasserstein distance (Panaretos & Zemel, 2019) to measure the discrepancy between generated and real distributions in a more stable manner. The kernelized Bures metric (Gilo et al., 2024) leverages the Bures metric within a reproducing kernel Hilbert space to compare source and target domain distributions. Existing analyses of Bures distance have treated it as a metric. We have amplified its advantages and, for the first time, extrapolated it to serve as an optimization objective for category discovery in open-world environments.

## 3 Methodology

### 3.1 Notation and Preliminaries of GCD

For each dataset, consider a labeled subset $\mathcal{D}_l = \{(\mathbf{x}_i^l, y_i^l)\} \subset \mathcal{X} \times \mathcal{Y}_l$ and an unlabeled subset $\mathcal{D}_u = \{(\mathbf{x}_i^u, y_i^u)\} \subset \mathcal{X} \times \mathcal{Y}_u$. Only known classes can be found in $\mathcal{D}_l$, while $\mathcal{D}_u$ encompasses known and novel classes, translating to $\mathcal{Y}_l = \mathcal{C}_{known}$ and $\mathcal{Y}_u = \mathcal{C}_{known} \cup \mathcal{C}_{novel}$. The task of models involves clustering on both the known and novel classes in $\mathcal{D}_u$. The number of novel classes

represented as $K_{novel}$ can be determined beforehand (Vaze et al., 2022b; Zhao et al., 2023). The functions $f(\cdot)$ and $g(\cdot)$ perform as the feature extractor and projection head, respectively. Both the feature $\mathbf{h}_i = f(\mathbf{x}_i)$ and the projected embedding $\mathbf{z}_i = g(\mathbf{h}_i)$ are under $\ell_2$-normalization. A ViT-style encoder (Dosovitskiy, 2020) provides one class token [cls] and $H \times W$ visual tokens per image.

## 3.2 TOKEN GEOMETRY

**Scope**. As shown in Figure 2, we first formalize the class-token space regulated by our method, keeping [cls] formation concise while retaining key mechanics for downstream spectral analysis. By *token geometry*, we mean the spectral/metric structure of class tokens (pairwise angles, covariance eigen-spectrum, effective rank, isotropy) computed over batches.

To simplify [cls]'s evidence aggregation, we summarize one attention step with one-head simplification (locally aggregating token evidence into a global summary). After linear projections $Q, K, V = XW_Q, XW_K, XW_V \in \mathbb{R}^{(1+HW) \times d_k}$ (or $d$ for $V$; $W_Q, W_K, W_V$ learnable, $d_k$ query/key dimension), let $q_c \in \mathbb{R}^{1 \times d_k}$ be [cls]'s row in $Q$. The class attention weights are:

$$\alpha_c = \text{softmax}\Big(\frac{q_c K^\top}{\sqrt{d_k}}\Big) \in \mathbb{R}^{1 \times (1+HW)}, \tag{1}$$

where $\alpha_c$ is a probability vector over all $(1 + HW)$ tokens. These normalized weights guide token evidence distribution and update [cls] residually. With the attention:

$$[\text{cls}]' = [\text{cls}] + \alpha_c V \in \mathbb{R}^{1 \times d}, \tag{2}$$

where [cls]$'$ is the updated [cls] and $V$ is the value matrix. Repeated across layers, Equations 1 and 2 defines [cls] for subsequent batch-scale geometric analysis. For GCD, token geometry governs two coupled goals: preserving intra-class completeness for known categories and enabling cohesive emergence of novel ones; in contrast, anisotropic, low-rank geometry under pseudo-label noise tends to induce spurious merges or over-fragmentations (Figure 1).

**Batch-scale token geometry**. To study geometry at the mini-batch scale, consider an unlabeled mini-batch $\mathcal{B}^u$ of size $B$ and stack per-image class tokens row-wise as

$$Z = \text{stack}([\text{cls}]_1, \ldots, [\text{cls}]_B) \in \mathbb{R}^{B \times d}, \qquad \Sigma_B = ZZ^\top \in \mathbb{R}^{B \times B}, \tag{3}$$

where $\text{stack}(\cdot)$ concatenates row vectors, $d$ is the embedding dimension of each class token, and $\Sigma_B$ is the Gram (Gatys et al., 2015) of class tokens. Under row $\ell_2$-normalization, $\text{tr}(\Sigma_B) = \|Z\|_F^2 = \sum_{i=1}^B \|Z_{i:}\|_2^2 \approx B$, and each entry $(\Sigma_B)_{ij} = \langle Z_{i:}, Z_{j:} \rangle$ reduces to a cosine similarity; thus the spectrum of $\Sigma_B$ compactly captures how class tokens co-occupy directions in feature space.

Since the feature Gram $Z^\top Z$ and sample Gram $ZZ^\top$ share identical nonzero singular spectra, we diagnose geometry via either the batch-scale $\Sigma_B$ or the global autocorrelation $\mathcal{A}$ (Sec. 3.4). A *uniform* eigen-spectrum signifies high effective rank and isotropic support, critical for GCD as the unlabeled pool mixes known and novel categories. Empirically, these isotropic manifolds provide richer discriminative capacity and tolerance to open-world uncertainty, stabilizing clustering updates against noise and class imbalance while improving class-number estimation.

## 3.3 BURES–ISOTROPY ALIGNMENT

**Motivation.** Token geometry strongly affects discovery quality, yet most GCD pipelines do not *explicitly* correct it in open-world settings. Building on our analysis above, we borrow the Bures metric (Bures, 1969; Uhlmann, 1976) from quantum information (a field that also values representation quality) and turn this metric into a simple optimization that matches GCD.

We first need a way to quantify "how far" a batch's class-token covariance is from an isotropic target; a natural choice is the Bures distance to identity

$$d_\text{B}^2(\Sigma_B, I) = \text{tr}(\Sigma_B) + B - 2\,\text{tr}\big(\Sigma_B^{1/2}\big), \tag{4}$$

where $\Sigma_B = ZZ^\top \in \mathbb{R}^{B \times B}$ is the batch Gram of stacked class tokens $Z \in \mathbb{R}^{B \times d}$, $B$ is batch size, and $I$ is the $B \times B$ identity; with row $\ell_2$-norms, the trace terms change little, so minimizing $d_{\mathrm{B}}^2$ mainly *increases* $\mathrm{tr}(\Sigma_B^{1/2})$ and thus spreads spectral energy.

To implement this without changing the training loop, we relate the square-root trace to a familiar surrogate that depends directly on $Z$:

$$\mathrm{tr}\big(\Sigma_B^{1/2}\big) = \sum_j \sqrt{\mu_j} = \sum_j s_j = \|Z\|_*, \tag{5}$$

where $\{\mu_j\}$ are eigenvalues of $\Sigma_B$ and $\{s_j\}$ are singular values of $Z$ and this identity converts the metric into the nuclear norm of stacked class tokens.

**Metric-to-loss.** With this link in place, aligning to isotropy becomes an extremely simple optimization on $Z$ under standard normalization:

$$\arg \min_Z d_{\mathrm{B}}^2(\Sigma_B, I) \;\equiv\; \arg \max_Z \|Z\|_*. \tag{6}$$

Intuitively, maximizing $\|Z\|_*$ lifts rank and homogenizes the spectrum, which raises entropy, reduces collapse, and stabilizes discovery in noisy unlabeled batches. We use either the metric itself or its nuclear-norm surrogate; both behave the same once row norms are stabilized.

$$\mathcal{L}_{\mathrm{BIA}} = d_{\mathrm{B}}^2(\Sigma_B, I) \qquad \text{or} \qquad \mathcal{L}_{\mathrm{BIA}}^{\mathrm{nuc}} = -\|Z\|_*. \tag{7}$$

This keeps the BIA architecture-agnostic and adds only a single scalar term to the base GCD objective:

$$\mathcal{L} = \mathcal{L}_{\mathrm{GCD}} + \lambda \mathcal{L}_{\mathrm{BIA}}. \tag{8}$$

In summary, token geometry is central to GCD but seldom corrected explicitly. By importing the Bures metric and recasting it as a nuclear-norm objective, BIA offers a direct way to restore isotropy in the class-token manifold and improve open-world discovery. As shown below, the implementation of BIA is extremely concise, with core code consisting of only two lines (provided in Section A).

### 3.4 BIA INCREASES VON NEUMANN ENTROPY

The autocorrelation matrix (Schölkopf & Smola, 2002) of the sample's token geometry is $\mathcal{A} \triangleq \sum_{i=1}^N \frac{1}{N} [\mathtt{cls}]_i [\mathtt{cls}]_i^\top = \mathbf{CLS}^\top \mathbf{CLS}/N$. We employ von Neumann entropy (Petz, 2001; Boes et al., 2019) to measure token geometry. This gives the advantage of focusing exclusively on the eigenvalues, allowing for graceful handling of eigenvalues that are extremely close to $0$. The von Neumann entropy can be expressed as $\hat{H}(\mathcal{A})$, representing the Shannon entropy (Shannon, 1948) of the eigenvalues of $\mathcal{A}$, with values ranging between $0$ and $\log d$. A larger $\hat{H}(\mathcal{A})$ indicates a greater token geometry capacity of the features. Von Neumann entropy is an effective measure for assessing the uniformity of distributions and managing extreme values.

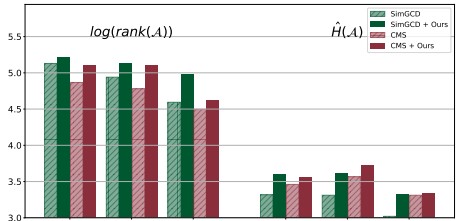

Figure 3: Comparison between $log(\mathrm{rank}(\mathcal{A}))$ and $\hat{H}(\mathcal{A})$. The count of the largest eigenvalues necessary to account for 99% of the total eigenvalue energy serves as a surrogate for the rank.

As illustrated in Figure 3, the incorporation of BIA results in a von Neumann entropy for the embeddings that is significantly higher than that of the original scheme. It is possible to relate von Neumann entropy to the rank of the [cls]. When $\mathcal{A}$ possesses uniformly distributed eigenvalues with full rank, the entropy is maximized. We clarify the connection between BIA and VNE. BIA serves as a local optimization objective for mini-batch Gram matrices. In contrast, VNE functions as a global diagnostic metric derived from the complete dataset's autocorrelation matrix. BIA enhances

Table 1: Experimental results on fine-grained datasets, evaluated *with* the $K$ for clustering.

| Method | CUB | | | Stanford Cars | | | FGVC Aircraft | | |
|---|---|---|---|---|---|---|---|---|---|
| | All | Old | New | All | Old | New | All | Old | New |
| *Clustering with the ground-truth number of classes K given* | | | | | | | | | |
| Agglomerative | 37.0 | 36.2 | 37.3 | 12.5 | 14.1 | 11.7 | 15.5 | 12.9 | 16.9 |
| RankStats+ | 33.3 | 51.6 | 24.2 | 28.3 | 61.8 | 12.1 | 26.9 | 36.4 | 22.2 |
| UNO+ | 35.1 | 49.0 | 28.1 | 35.5 | 70.5 | 18.6 | 40.3 | 56.4 | 32.2 |
| ORCA | 35.3 | 45.6 | 30.2 | 23.5 | 50.1 | 10.7 | 22.0 | 31.8 | 17.1 |
| GCD | 51.3 | 56.6 | 48.7 | 39.0 | 57.6 | 29.9 | 45.0 | 41.1 | 46.9 |
| ProtoGCD | 63.2 | 68.5 | 60.5 | 53.8 | 73.7 | 44.2 | 56.8 | 62.5 | 53.9 |
| PrCAL | 62.9 | 64.4 | 62.1 | 50.2 | 70.1 | 40.6 | 52.2 | 52.2 | 52.3 |
| ActiveGCD | 66.6 | 66.5 | 66.7 | 48.4 | 57.7 | 39.3 | 53.7 | 51.5 | 56.0 |
| PIM | 62.7 | 75.7 | 56.2 | 43.1 | 66.9 | 31.6 | - | - | - |
| SelEx | 78.7 | **81.3** | 77.5 | 55.9 | 76.9 | 45.8 | 60.8 | **70.3** | 56.2 |
| + Ours | **80.6** | 81.0 | **80.4** | 57.0 | **77.3** | 47.2 | **61.8** | 68.2 | **59.2** |
| | **+1.9** | -0.3 | **+2.9** | **+1.1** | **+0.4** | **+1.4** | **+1.0** | -2.1 | **+3.0** |
| SimGCD | 60.7 | 65.6 | 57.7 | 51.2 | 69.4 | 42.4 | 54.0 | 58.8 | 51.5 |
| + Ours | 62.1 | 65.8 | 60.3 | 52.3 | 70.0 | 43.7 | 55.1 | 58.9 | 53.1 |
| | **+1.4** | **+0.2** | **+2.6** | **+1.1** | **+0.6** | **+1.3** | **+1.1** | **+0.1** | **+1.6** |
| CMS | 67.1 | 74.9 | 63.2 | 56.7 | 76.8 | 37.5 | 53.6 | 60.3 | 47.0 |
| + Ours | 71.1 | 74.1 | 66.9 | 57.4 | 79.4 | 36.2 | 55.7 | 63.7 | 47.9 |
| | **+4.0** | -0.8 | **+3.7** | **+0.7** | **+2.6** | -1.3 | **+2.1** | **+3.4** | **+0.9** |
| SPTNet | 62.0 | 69.2 | 56.0 | 56.2 | 70.3 | 46.6 | 51.6 | 60.7 | 45.9 |
| + Ours | 63.3 | 70.7 | 59.6 | **58.8** | 75.4 | **50.8** | 54.7 | 65.3 | 48.5 |
| | **+1.3** | **+1.5** | **+3.6** | **+2.6** | **+5.1** | **+4.2** | **+3.1** | **+4.5** | **+2.6** |
| **Avg. △** | **+2.2** | **+0.2** | **+3.2** | **+1.4** | **+2.2** | **+1.1** | **+1.8** | **+1.5** | **+2.0** |

the feature geometry directly, whereas VNE assesses the resulting improvements by evaluating spectral uniformity and effective rank.

# 4 EXPERIMENTS

## 4.1 SETUP

**Benchmarks**. BIA is evaluated on coarse- and fine-grained benchmarks. These include two conventional datasets, CIFAR100 (Krizhevsky & Hinton, 2009) and ImageNet100 (Geirhos et al., 2019), and four fine-grained datasets, CUB-200-2011 (Wah et al., 2011), Stanford Cars (Krause et al., 2013), FGVC Aircraft (Maji et al., 2013), and Herbarium19 (Tan et al., 2019). To segregate target classes into sets of known and unknown, we adhere to the splits defined by the SSB (Vaze et al., 2022a) when working with CUB, Stanford Cars, and FGVC Aircraft. The splits from the previous study (Vaze et al., 2022b) are employed for the remaining datasets, we designate 80% of the classes as known under the CIFAR100 benchmark. For the rest of the benchmarks, the proportion of known classes stands at 50%. Our labeled set $\mathcal{D}_l$, comprises 50% images from the known classes for all benchmarks.

**Evaluation Protocols**. We assess BIA's effectiveness via a two-step process. First, we cluster the complete collection of images defined as $\mathcal{D}$. Then, we measure the accuracy on the set $\mathcal{D}_u$. In line with previous research (Vaze et al., 2022b), accuracy is determined by comparing the assignments to the actual labels using the Hungarian optimal matching (Kuhn, 1955). This method bases the match on the number of instances that intersect between each pair of classes. Instances that do not belong to any pair, *i.e.*, unpaired classes, are viewed as incorrect predictions. On the other hand, instances belonging to the most abundant class within each ground-truth cluster are taken as correct for accuracy calculations. We present the accuracy for all unlabeled data, and the accuracy is classified as old/known and new/novel, respectively. The accuracy using the estimated number of classes and the ground-truth $K$ are reported. This allows us to compare BIA with previous studies that have assumed the availability of the $K$ during the evaluation phase.

**Implementation Details**. The purpose of BIA is to empower existing GCD schemes to improve the completeness of representation. We closely adhere to their initial implementation details for an effective comparison. We use a pre-trained DINO ViT-B/16 (Caron et al., 2021; Dosovitskiy, 2020),as our image encoder along with a projection head, an approach consistent with existing methods (Vaze et al., 2022b; Zhang et al., 2023; Pu et al., 2023). All experiments are performed on four NVIDIA

Table 2: Experimental results on coarse- and fine-grained datasets, evaluated *with* the $K$ for clustering.

| Method | CIFAR100 | | | ImageNet100 | | | Herbarium 19 | | |
|---|---|---|---|---|---|---|---|---|---|
| | All | Old | New | All | Old | New | All | Old | New |
| *Clustering with the ground-truth number of classes $K$ given* | | | | | | | | | |
| Agglomerative | 56.9 | 56.6 | 57.5 | 73.1 | 77.9 | 70.6 | 14.4 | 14.6 | 14.4 |
| RankStats+ | 58.2 | 77.6 | 19.3 | 37.1 | 61.6 | 24.8 | 27.9 | 55.8 | 12.8 |
| UNO+ | 69.5 | 80.6 | 47.2 | 70.3 | 95.0 | 57.9 | 28.3 | 53.7 | 14.7 |
| ORCA | 69.0 | 77.4 | 52.0 | 73.5 | 92.6 | 63.9 | 20.9 | 30.9 | 15.5 |
| GCD | 73.0 | 76.2 | 66.5 | 74.1 | 89.8 | 66.3 | 35.4 | 51.0 | 27.0 |
| ProtoGCD | 81.9 | 82.9 | 80.0 | 84.0 | 92.2 | 79.9 | 44.5 | 59.4 | 36.5 |
| PrCAL | 81.2 | 84.2 | 75.3 | 83.1 | 92.7 | 78.3 | 37.0 | 52.0 | 28.9 |
| ActiveGCD | 71.3 | 75.7 | 66.8 | 83.3 | 90.2 | 76.5 | - | - | - |
| PIM | 78.3 | 84.2 | 66.5 | 83.1 | 95.3 | 77.0 | 42.3 | 56.1 | 34.8 |
| SelEx | 80.0 | 84.8 | 70.4 | 82.3 | 93.9 | 76.5 | 36.2 | 46.0 | 30.9 |
| + Ours | 80.7 | 84.3 | 72.1 | 82.8 | 94.1 | 77.8 | 36.8 | 47.5 | 31.0 |
| | +0.7 | -0.5 | +1.7 | +0.5 | +0.2 | +1.3 | +0.6 | +1.5 | +0.1 |
| SimGCD | 80.1 | 81.5 | 77.2 | 83.3 | 92.1 | 78.9 | 44.7 | 57.4 | 37.9 |
| + Ours | 80.2 | 81.5 | **77.5** | **86.7** | 93.1 | **83.6** | **45.6** | 57.8 | **39.0** |
| | +0.1 | +0.0 | +0.3 | +3.4 | +1.0 | +4.7 | +0.9 | +0.4 | +1.1 |
| CMS | 79.5 | 85.4 | 67.7 | 83.0 | **95.6** | 76.6 | 36.5 | 55.4 | 26.4 |
| + Ours | 79.0 | **85.5** | 66.1 | 84.8 | **95.6** | 79.5 | 36.3 | 56.5 | 25.4 |
| | -0.5 | +0.1 | -1.6 | +1.8 | +0.0 | +2.9 | -0.2 | +1.1 | -1.0 |
| SPTNet | 81.1 | 84.3 | 75.0 | 85.3 | 93.1 | 81.3 | 43.5 | **58.9** | 35.6 |
| + Ours | **82.1** | 84.8 | 76.2 | 85.4 | 93.4 | 81.3 | 44.2 | **58.9** | 36.3 |
| | +1.0 | +0.5 | +1.2 | +0.1 | +0.3 | +0.0 | +0.7 | +0.0 | +0.7 |
| **Avg. △** | **+0.3** | **+0.1** | **+0.4** | **+1.5** | **+0.4** | **+2.2** | **+0.5** | **+0.7** | **+0.2** |

Table 3: GCD Accuracy on coarse- and fine-grained datasets, evaluated *without* the $K$ for clustering.

| Method | CIFAR100 | | | ImageNet100 | | | CUB | | | Stanford Cars | | | FGVC Aircraft | | | Herbarium 19 | | |
|---|---|---|---|---|---|---|---|---|---|---|---|---|---|---|---|---|---|---|
| | All | Old | New | All | Old | New | All | Old | New | All | Old | New | All | Old | New | All | Old | New |
| *Clustering without the ground-truth number of classes $K$ given* | | | | | | | | | | | | | | | | | | |
| Agglomerative | 56.9 | 56.6 | 57.5 | 72.2 | 77.8 | 69.4 | 35.7 | 33.3 | 36.9 | 10.8 | 10.6 | 10.9 | 14.1 | 10.3 | 16.0 | 13.9 | 13.6 | 14.1 |
| GCD | 70.8 | 77.6 | 57.0 | 77.9 | 91.1 | 71.3 | 51.1 | 56.4 | 48.4 | 39.1 | 58.6 | 29.7 | - | - | - | 37.2 | 51.7 | 29.4 |
| GPC | 75.4 | 84.6 | 60.1 | 75.3 | 93.4 | 66.7 | 52.0 | 55.5 | 47.5 | 38.2 | 58.9 | 27.4 | 43.3 | 40.7 | 44.8 | 36.5 | 51.7 | 27.9 |
| PIM | 75.6 | 81.6 | 63.6 | 83.0 | 95.3 | 76.9 | 62.0 | **75.7** | 55.1 | 42.4 | 65.3 | 31.3 | - | - | - | **42.0** | 55.5 | **34.7** |
| CMS | 77.8 | 84.0 | 65.3 | 83.4 | 95.6 | 77.3 | 66.2 | 69.7 | 64.4 | 51.8 | **72.9** | 31.3 | 52.3 | 58.9 | 45.8 | 38.5 | **57.3** | 28.4 |
| + Ours | **79.5** | **84.7** | **69.1** | **84.3** | **95.7** | **78.8** | **68.7** | 74.1 | **66.0** | **52.5** | 72.7 | **32.9** | **53.4** | **60.1** | **46.7** | 38.0 | 56.9 | 27.9 |
| **Avg. △** | **+1.7** | **+0.7** | **+3.8** | **+0.9** | **+0.1** | **+1.5** | **+2.5** | **+4.4** | **+1.6** | **+0.7** | -0.2 | **+1.6** | **+1.1** | **+1.2** | **+0.9** | -0.5 | -0.4 | -0.5 |

RTX 4090 GPUs. **We follow the original training parameter details of each scheme to illustrate the generality and applicability of BIA**.

## 4.2 MAIN RESULTS

**Evaluation on GCD.** As shown in Tables 1, 2 and 3, BIA brings consistent and notable gains across all evaluated GCD methods and datasets, under both known and unknown class number settings. Key findings are as follows: ❶ *Compatibility.* BIA improves all baselines including SimGCD, CMS, SPTNet, and SelEx without any architectural changes. For example, on CUB with known class number, BIA enhances SimGCD by 2.6% on novel classes and CMS by 4.0% on all classes. On ImageNet100, it improves SimGCD by 3.4% in the all setting and boosts CMS by 2.9% on novel classes. These results highlight BIA's strong generalization across frameworks and confirm its plug-and-play compatibility. ❷ *Generality.* BIA yields stable gains on both coarse-grained datasets like CIFAR100 and ImageNet100 and fine-grained ones like CUB and Cars. Notably, on CUB, BIA improves CMS (known) by 4.4% under unknown class number settings. Average improvements on novel classes range from about 0.9% to 3.8% across datasets, demonstrating the robustness to domain complexity and label granularity.

**Ablation study.** The only hyperparameter of BIA is the coefficient $\lambda$ of the loss. To gain a deeper understanding of the correlation between the degree of maximum token manifold capacity and the dimensionality $D$ of the features, we conducted an ablation experiment on it, as shown

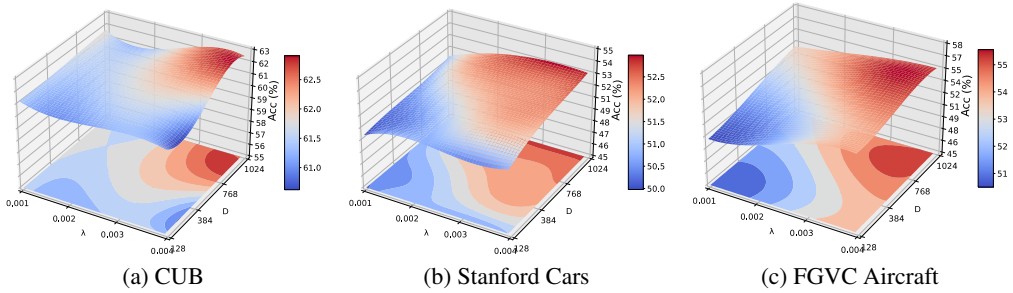

(a) CUB        (b) Stanford Cars        (c) FGVC Aircraft

Figure 4: Hyperparameter sensitivity of the degree of BIA ($\lambda$) and features dimensionality ($D$).

in Figure 4. It can be observed that BIA is not sensitive to hyperparameters and can uniformly enhance clustering accuracy. A more thought-provoking finding is that directly reducing $D$ to avoid dimensionality collapse is suboptimal. The reason is that each dimension of the manifold contributes to the representation, and a reduction in $D$ will directly lead to a loss of information. Even with BIA, it is impossible to make the representation complete. An appropriate number of dimensions enriches the representation while using BIA to prevent dimensionality collapse, which can maximize the model's performance enhancement.

## 5   Hierarchical Analysis of Why BIA is Effective

We conduct a comprehensive analysis from multiple dimensions: 1) eigenvalue distribution and Frobenius norm, 2) estimation of embedded space distribution, 3) dimensional collapse, and 4) comparison with similar schemes, to understand the necessity and effectiveness of BIA for GCD.

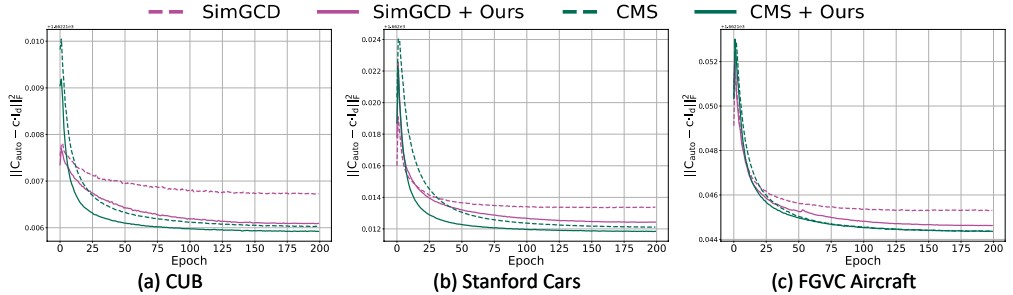

(a) CUB        (b) Stanford Cars        (c) FGVC Aircraft

Figure 5: The Frobenius norm $\|\mathcal{A} - c \cdot I_d\|_F^2$ on three fine-grained benchmarks.

### 5.1   BIA Homogenizes Eigenvalue Distribution and Reduces Frobenius Norm

The autocorrelation matrix of the test sample token geometry is denoted as $\mathcal{A}$. Given $\|[\texttt{cls}]_i\|_2 = 1$ and $\mathcal{A} \geq 0$, it follows that $\sum_j \lambda_j = 1$ and $\forall_j \lambda_j \geq 0$ (Parkhi et al., 2015; Liu et al., 2017; Mettes et al., 2019), where $\{\lambda_j\}$ are the eigenvalues of $\mathcal{A}$. Under ideal conditions, where $\mathcal{A} \to c \cdot I_d$, the eigenvalue distribution of $\mathcal{A}$ becomes uniform, $\mathbf{z}$ uncorrelated (Cogswell et al., 2015), full-rank (Hua et al., 2021), and isotropic (Vershynin, 2018). $\mathcal{A}$ is linked to various representation characteristics. The Frobenius norm (Ma et al., 1994; Peng et al., 2016), extensively studied in self-supervised learning methods (Cogswell et al., 2015; Xiong et al., 2016; Choi & Rhee, 2019; Zbontar et al., 2021), measures whether the representation depends on a few dimensions. A smaller Frobenius norm indicates a larger manifold capacity. We applied singular value decomposition (SVD) (Golub & Reinsch, 1971) to the autocorrelation matrix of the feature embeddings, plotting the first 200 singular values in Figure 6 and visualizing the Frobenius norm $|\mathcal{A} - c \cdot I_d|_F^2$ in Figure 5. Compared to SimGCD and CMS, BIA achieves a more uniform and stable eigenvalue distribution.

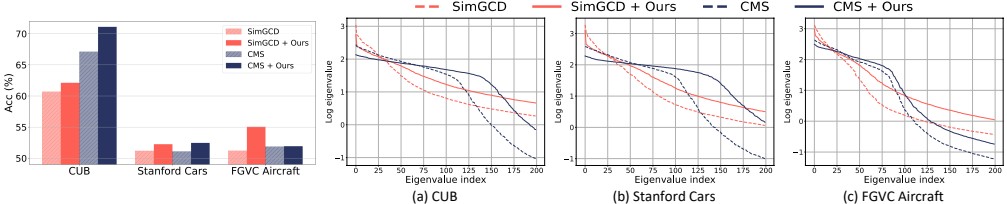

Figure 6: BIA effectively mitigates dimensional collapse by providing a more uniform eigenvalue distribution and improves the clustering accuracy.

## 5.2 BIA UNRAVELS DIMENSIONAL COLLAPSE.

We further explored the relationship between the accuracy and eigenvalues of GCD, respectively, and dimensional collapse, as shown in Figure 6 and our findings are as follows: (1) Feature Completeness and Clustering Accuracy: Complete features improve intra-class representations, which enhances clustering accuracy by providing richer, higher manifold capacity. (2) BIA's Impact: BIA increases manifold capacity, leading to higher singular values and more accurate clustering by better approximating the true distribution. (3) Dimension Collapse and Limitations of CMS/SimGCD: CMS and SimGCD operate in lower-dimensional spaces, limiting manifold capacity and causing incomplete representations (Caron et al., 2020; Shi et al., 2023). Dimension collapse results in oversimplified models, while BIA maximizes intra-class completeness for better decision boundaries. This breakdown highlights how BIA addresses limitations in existing methods by optimizing the manifold capacity and the richness of intra-class representations, leading to improved model performance.

## 5.3 BIA PROVIDES ACCURATE DISTRIBUTION ESTIMATION

We present the gap between BIA and SOTAs in estimating the number of clusters in Table 4. By leveraging CMS, which requires no specific hyperparameters to estimate $K$, our optimization target becomes $\mathcal{L}_{CMS} + \mathcal{L}_{BIA}$. Results demonstrate significant improvement with BIA incorporated into the CMS framework. Notably, on the complex and diverse ImageNet100 dataset,

Table 4: Estimated number and error rate of $K$.

| Method | CIFAR100 | | ImageNet100 | | CUB | | Stanford Cars | | FGVC Aircraft | |
|---|---|---|---|---|---|---|---|---|---|---|
| | K | Err(%) | K | Err(%) | K | Err(%) | K | Err(%) | K | Err(%) |
| Ground truth | 100 | - | 100 | - | 200 | - | 196 | - | 100 | - |
| GCD | 100 | 0 | 109 | 9 | 231 | 15.5 | 230 | 17.3 | - | - |
| DCCL | 146 | 46 | 129 | 29 | 172 | 9 | 192 | 0.02 | - | - |
| PIM | 95 | 5 | 102 | 2 | 227 | 13.5 | 169 | 13.8 | - | - |
| GPC | 100 | 0 | 103 | 3 | 212 | 6 | 201 | 0.03 | - | - |
| CMS† | 94 | 6 | 98 | 2 | 176 | 12 | 149 | 23.9 | | |
| + Ours | 96 | 4 | 100 | 0 | 180 | 10 | 159 | 18.9 | 89 | 11 |

our method achieves a 100% correct estimation rate. The improvement in estimating the number of clusters highlights the importance of representation completeness, enabling better capture of intra-class nuances and sharper inter-class separation.

## 5.4 COMPARISON WITH OTHER ISOTROPIC DISTRIBUTION SCHEMES

From the perspective of motivation and self-supervised learning based on Isotropic Distribution, to which BIA is similar, we chose representatives CorInfoMax (Ozsoy et al., 2022) and VICReg (Bardes et al., 2022) as challengers. As shown in Table 5, in the context of GCD, VICReg, while promoting variance and reducing covariance, does not explicitly focus on maximizing intra-class representation completeness, which is crucial for distinguishing fine-grained categories. CorInfoMax, on the other hand, pri-

Table 5: Comparison on accuracy in GCD with representative isotropic feature distribution schemes.

| Method | CUB | | | Stanford Cars | | | FGVC Aircraft | | | Average | | |
|---|---|---|---|---|---|---|---|---|---|---|---|---|
| | All | Old | New | All | Old | New | All | Old | New | All | Old | New |
| SimGCD | 60.7 | 65.6 | 57.7 | 51.2 | 69.4 | 42.4 | 54.0 | 58.8 | 51.5 | 55.3 | 64.6 | 50.5 |
| +CorInfoMax | 60.7 | 64.8 | 58.6 | 50.0 | 67.4 | 41.6 | 54.4 | **59.0** | 52.1 | 55.0 | 63.7 | 50.8 |
| +VICReg | 61.1 | **66.0** | 58.1 | 52.0 | 68.6 | **44.1** | 54.6 | 56.2 | **53.8** | 55.9 | 63.6 | 52.0 |
| +Ours | **62.1** | 65.8 | **60.3** | **52.3** | **70.0** | 43.7 | **55.1** | 58.9 | 53.1 | **56.5** | **64.9** | **52.4** |
| CMS | 67.1 | 74.9 | 63.2 | 56.7 | 76.8 | 37.5 | 53.6 | 60.3 | 47.0 | 59.1 | 70.7 | 49.2 |
| +CorInfoMax | 65.7 | 76.4 | 58.7 | 55.8 | 73.1 | 39.2 | 52.4 | 61.9 | 42.8 | 58.0 | 70.5 | 46.9 |
| +VICReg | 68.3 | **78.1** | 55.0 | **57.8** | 76.7 | **39.7** | 55.2 | **65.2** | 45.1 | 60.4 | **73.3** | 46.6 |
| +Ours | **71.1** | 74.1 | **66.9** | 57.4 | **79.4** | 36.2 | **55.7** | 63.7 | **47.9** | **61.4** | 72.4 | **50.3** |

marily maximizes mutual information but does not explicitly prevent dimensional collapse or ensure richer intra-class representations. As a result, both methods struggle to capture the full complexity of the structure of the data, limiting their effectiveness in accurately discovering novel categories.

# 6 CONCLUSION

We introduce Bures-Isotropy Alignment, a simple yet powerful method for enhancing generalized category discovery, which identifies known classes and discovers novel ones from unlabeled data containing both categories as a core demand of open-world learning. BIA effectively addresses the limitation of traditional GCD methods, which often sacrifice representation quality for compact clustering (causing dimensional collapse, distorted eigen-structure), and fragile feature distributions that hinder category discovery. Inspired by quantum information science's pursuit of representational completeness, BIA optimizes class-token covariance toward an isotropic prior by minimizing the Bures distance. Under a mild trace constraint, this equals maximizing the nuclear norm of class tokens (no model architecture changes), restoring the over-flattened feature space to ensure complete, rich intra-class representations while preserving inter-class separability. BIA significantly boosts accuracy (All/Old/New) and class-number estimation of GCD baselines with negligible computational cost. By restoring feature geometry, BIA unlocks the model's full potential, meets GCD's needs, and serves as an effective tool for more adaptable machine learning models in open-world scenarios.

## ACKNOWLEDGMENTS

The work described in this paper is supported by grants from HKU Startup Fund and HKU Seed Fund for Basic Research

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

## USE OF LLMS

We use large language models (LLMs) solely for language polishing of the final manuscript (correcting grammatical errors and refining expression). The models play no part in conceptualization, experimental design, theoretical analysis, or any substantive writing. All scientific viewpoints and results remain our sole responsibility.

## ETHICS STATEMENT

This study strictly adheres to the ethical guidelines and submission requirements of ICLR. The data and code used are legally sourced, with no unauthorized usage. The experimental code is either independently developed or reasonably modified based on open-source projects, in compliance with intellectual property regulations, and is prepared for public release as required. All authors declare no relevant conflicts of interest, and the research conclusions are not unduly influenced. The entire work fully meets the academic ethics and compliance standards of ICLR

## REPRODUCIBILITY STATEMENT

The code is provided in the supplementary materials to replicate the empirical results.

## A  PYTORCH-STYLE IMPLEMENTATION OF BIA

```python
def forward(self, x_unlabel, loss):
    f_cls_unlabel = self.featurizer(x_unlabel)[:,0] # get class token
    z_cls_unlabel = self.projector(f_cls_unlabel) # get feature embedding
    _,s,_ = torch.svd(z_cls_unlabel) # singular value decomposition
    loss -= self.lambda * torch.sum(s) # BIA
    return loss
```

## B  DETAILS OF OPTIMIZATION OBJECTIVE OF GCD

The existing GCD proposals are all proposed for compact clustering. Summarizing the optimization objectives of mainstream schemes GCD (Vaze et al., 2022b), CMS (Choi et al., 2024) and SimGCD (Wen et al., 2023), it can be observed that they are based on contrastive learning or prototype learning to significantly reduce the distance between potentially similar samples in the feature space.

### B.1  GCD

The pioneering work (Vaze et al., 2022b) divided the mini-batch $\mathcal{B}$ into labeled $\mathcal{B}^l$ and unlabeled $\mathcal{B}^u$, using supervised (Khosla et al., 2020) contrastive learning $\mathcal{L}_{\text{GCD}}^l = -\frac{1}{|\mathcal{B}^l|} \sum_{i \in \mathcal{B}^l} \frac{1}{|\mathcal{B}^l(i)|} \sum_{j \in \mathcal{B}^l(i)} \log \frac{\exp(\mathbf{z}_i^\top \mathbf{z}_j'/\tau)}{\sum_{n \neq i} \exp(\mathbf{z}_i^\top \mathbf{z}_n'/\tau)}$, and self-supervised (Chen et al., 2020) contrastive learning $\mathcal{L}_{\text{GCD}}^u = -\frac{1}{|\mathcal{B}|} \sum_{i \in \mathcal{B}} \log \frac{\exp(\mathbf{z}_i^\top \mathbf{z}_i'/\tau)}{\sum_{n \neq i} \exp(\mathbf{z}_i^\top \mathbf{z}_n'/\tau)}$ and balancing them using coefficients $\lambda$: $\mathcal{L}_{\text{GCD}} = (1-\lambda)\mathcal{L}_{\text{GCD}}^u + \lambda\mathcal{L}_{\text{GCD}}^l$, where $\mathcal{B}^l(i)$ represents the collection of samples with the same label as $i$. The $\mathbf{z}$ and $\mathbf{z}'$ are augmented from two different views, and the $\tau$ is the temperature.

### B.2  CMS

CMS (Choi et al., 2024) and GCD introduces similar supervised and self-supervised contrastive learning. The difference is that CMS introduced mean-shift into unsupervised learning. For the $i$-th sample, CMS collects the feature set $\mathcal{V} = \{\mathbf{z}_i\}_{i=1}^N$ of training samples and calculates the k-nearest neighbours $\mathcal{N}(\mathbf{z}_i) = \{\mathbf{z}_i\} \cup \text{argmax}_{\mathbf{z}_j \in \mathcal{V}}^k \mathbf{z}_i \cdot \mathbf{z}_j$, where $\text{argmax}_{s \in \mathcal{S}}^k(\cdot)$ returns a subset of the top-$k$ items. By aggregating neighbor embeddings with weight kernel $\varphi(\cdot)$, it obtains the new

embedded representation of samples after mean-shift: $\hat{\mathbf{z}}_i = \frac{\sum_{\mathbf{z}_j \in \mathcal{N}(\mathbf{z}_i)} \varphi(\mathbf{z}_j - \mathbf{z}_i)\mathbf{z}_j}{\left\| \sum_{\mathbf{z}_j \in \mathcal{N}(\mathbf{z}_i)} \varphi(\mathbf{z}_j - \mathbf{z}_i)\mathbf{z}_j \right\|}$. $\mathcal{L}_{\text{CMS}}$ and $\mathcal{L}_{\text{GCD}}$ are formally approximate.

### B.3 SIMGCD

SimGCD (Wen et al., 2023) constructs a prototype classifier $\mathbf{C} = \{\mathbf{c}_1, \cdots, \mathbf{c}_{K_{\text{known}}+K_{\text{novel}}}\}$ for both known and unknown classes. It obtains the posterior probability $\mathbf{p}_i^{(k)} = \frac{\exp(\mathbf{h}_i^\top \mathbf{c}_k)/\tau}{\sum_{k'} \exp(\mathbf{h}_i^\top \mathbf{c}_k')/\tau}$ in a similar way to FixMatch and uses cross-entropy loss $\mathcal{L}_{\text{SimGCD}}^l = \frac{1}{|\mathcal{B}^l|} \sum_{i \in \mathcal{B}^l} \ell(y_i, \mathbf{p}_i)$ on labeled samples. Self-distillation and entropy regularization $\mathcal{L}_{\text{SimGCD}}^u = \frac{1}{|\mathcal{B}|} \ell(\mathbf{p}_i', \mathbf{p}_i) - \lambda_e H(\frac{1}{2|\mathcal{B}|} \sum_{i \in \mathcal{B}} (\mathbf{p}_i + \mathbf{p}_i'))$ are performed using augmented samples with probability $\mathbf{p}_i'$.

## C PROOFS OF THEOREM

**Lemma 1.** *Given non-negative values $p_i$ such that $\sum_{i=1}^n p_i = 1$, the entropy function $H(p_1, \ldots, p_n) = -\sum_{i=1}^n p_i \log p_i$ is strictly concave. Furthermore, it is upper-bounded by $\log n$, as demonstrated by the inequality:*

$$\log n = H(1/n, \ldots, 1/n) \geq H(p_1, \ldots, p_n) \geq 0. \tag{9}$$

**Proof C.1.** *Refer to Section D.1 in (Marshall, 1979).*

**Lemma 2.** *The Kullback-Leibler (KL) divergence between two zero-mean, $d$-dimensional multivariate Gaussian distributions can be formulated as follows:*

$$D_{\text{KL}}(\mathcal{N}(0, \boldsymbol{\Sigma}_1) \| \mathcal{N}(0, \boldsymbol{\Sigma}_2))$$
$$= \frac{1}{2} \left[ \text{tr}(\boldsymbol{\Sigma}_2^{-1} \boldsymbol{\Sigma}_1) - d + \log \frac{|\boldsymbol{\Sigma}_2|}{|\boldsymbol{\Sigma}_1|} \right]. \tag{10}$$

**Proof C.2.** *Refer to Section 9 in (Duchi, 2007).*

**Theorem 1.** *For a given [cls] autocorrelation matrix $\mathcal{A} = \mathbf{CLS}^\top \mathbf{CLS}/N \in \mathbb{R}^{d \times d}$ of rank $k$ ($\leq d$),*

$$\log(\text{rank}(\mathcal{A})) \geq \hat{H}(\mathcal{A}), \tag{11}$$

*where equality holds if the eigenvalues of $\mathcal{A}$ are uniformly distributed with $\forall_{j=1}^k \lambda_j = 1/k$ and $\forall_{j=k+1}^d \lambda_j = 0$.*

**Proof C.3.** *We rely on the property that the sum of eigenvalues equals 1 (see belows for the detailed proof).*

$$\log(\text{rank}(\mathcal{A})) = \log(k) \tag{12}$$
$$\geq H(\lambda_1, \ldots, \lambda_k) \quad \text{(by Lemma 1)} \tag{13}$$
$$= -\sum_{j=1}^k \lambda_j \log \lambda_j \tag{14}$$
$$= -\sum_{j=1}^d \lambda_j \log \lambda_j \tag{15}$$
$$= \hat{H}(\mathcal{A}). \tag{16}$$

*According to Lemma 1, the inequality equation 13 attains equality if and only if $\lambda_j = \frac{1}{k}$ for all $j = 1, 2, \ldots, k$. Equation equation 15 adheres to the convention that $0 \log 0 = 0$, as per the definition in (Thomas & Joy, 2006).*

Here we provide the detailed proof that the sum of eigenvalues of the autocorrelation matrix $\mathcal{A}$ is 1.

Suppose we have a set of $n$ normalized vectors $\mathbf{v}_1, \mathbf{v}_2, \ldots, \mathbf{v}_n \in \mathbb{R}^d$, where the $\ell_2$-norm of each vector is 1, i.e., $\|\mathbf{v}_i\|_2 = 1$ for all $i$. The autocorrelation matrix $\mathcal{A}$ is defined as the average of the outer products:

$$\mathcal{A} = \frac{1}{n} \sum_{i=1}^n \mathbf{v}_i \mathbf{v}_i^\top. \tag{17}$$

We seek to show that $\sum_{j=1}^{d} \lambda_j = 1$, where $\{\lambda_j\}$ are the eigenvalues of $\mathcal{A}$. Recall that the trace of a matrix is equal to the sum of its eigenvalues, i.e., $\mathrm{tr}(\mathcal{A}) = \sum_{j=1}^{d} \lambda_j$.

By the linearity of the trace operator, we have:

$$\mathrm{tr}(\mathcal{A}) = \mathrm{tr}\left(\frac{1}{n}\sum_{i=1}^{n}\mathbf{v}_i\mathbf{v}_i^{\top}\right) = \frac{1}{n}\sum_{i=1}^{n}\mathrm{tr}(\mathbf{v}_i\mathbf{v}_i^{\top}). \tag{18}$$

Using the cyclic property of the trace, specifically $\mathrm{tr}(\mathbf{x}\mathbf{y}^{\top}) = \mathbf{x}^{\top}\mathbf{y}$, we obtain:

$$\mathrm{tr}(\mathbf{v}_i\mathbf{v}_i^{\top}) = \mathbf{v}_i^{\top}\mathbf{v}_i = \|\mathbf{v}_i\|_2^2. \tag{19}$$

Since the vectors are normalized ($\|\mathbf{v}_i\|_2 = 1$), it follows that $\|\mathbf{v}_i\|_2^2 = 1$. Substituting this back into the trace equation:

$$\mathrm{tr}(\mathcal{A}) = \frac{1}{n}\sum_{i=1}^{n}1 = \frac{1}{n}\cdot n = 1. \tag{20}$$

Thus, the sum of the eigenvalues of $\mathcal{A}$ is exactly 1.

## C.1 BIA's surrogate under a trace constraint

For completeness, we justify the equivalence between the Bures-based loss and the nuclear-norm surrogate used in our implementation. Here we provide a proof sketch and defer further discussion to this appendix.

Let $Z \in \mathbb{R}^{B \times d}$ be the matrix of class tokens in a mini-batch after LayerNorm, and define the batch Gram $\Sigma_B = ZZ^{\top} \in \mathbb{R}^{B \times B}$. Since LayerNorm makes each row approximately unit-norm, we have

$$\|z_i\|_2 \approx 1 \quad \Longrightarrow \quad \mathrm{tr}(\Sigma_B) = \mathrm{tr}(ZZ^{\top}) = \|Z\|_F^2 \approx B. \tag{21}$$

Thus, throughout training the trace of $\Sigma_B$ is well concentrated around the constant $B$. In other words, the eigenvalues $\{\mu_j\}_{j=1}^{B}$ of $\Sigma_B$ lie on the simplex $\sum_j \mu_j \approx B$.

The squared Bures distance between $\Sigma_B$ and the identity $I_B$ is

$$d_B^2(\Sigma_B, I_B) = \mathrm{tr}(\Sigma_B) + \mathrm{tr}(I_B) - 2\,\mathrm{tr}\big(\Sigma_B^{1/2}\big) = \mathrm{tr}(\Sigma_B) + B - 2\,\mathrm{tr}\big(\Sigma_B^{1/2}\big). \tag{22}$$

Combining equation 21 and equation 22, and treating $\mathrm{tr}(\Sigma_B)$ as approximately constant, we obtain

$$d_B^2(\Sigma_B, I_B) \approx 2B - 2\,\mathrm{tr}\big(\Sigma_B^{1/2}\big). \tag{23}$$

Consequently, minimizing the Bures loss is (up to an additive constant) equivalent to maximizing $\mathrm{tr}(\Sigma_B^{1/2})$.

We now relate $\mathrm{tr}(\Sigma_B^{1/2})$ to the nuclear norm of $Z$. Let $\{\mu_j\}_{j=1}^{B}$ denote the eigenvalues of $\Sigma_B$ and $\{s_j(Z)\}$ the singular values of $Z$. By construction $\Sigma_B = ZZ^{\top}$, so its non-zero eigenvalues coincide with the squared singular values of $Z$:

$$\mu_j = s_j(Z)^2 \quad \text{for all non-zero modes } j. \tag{24}$$

Therefore

$$\mathrm{tr}\big(\Sigma_B^{1/2}\big) = \sum_j \sqrt{\mu_j} = \sum_j s_j(Z) = \|Z\|_*, \tag{25}$$

the nuclear norm of $Z$.

Putting equation 23 and equation 25 together, we obtain the following lemma.

**Lemma 3** (Bures–nuclear norm equivalence). *Let $Z \in \mathbb{R}^{B \times d}$ and $\Sigma_B = ZZ^{\top}$. Suppose that $\mathrm{tr}(\Sigma_B)$ is (approximately) constant, as induced by LayerNorm or $\ell_2$ normalization on rows of $Z$. Then any minimizer of the Bures distance $d_B^2(\Sigma_B, I_B)$ is a maximizer of the nuclear norm $\|Z\|_*$, and conversely, up to an additive constant independent of $Z$.*

**Proof C.4** (Proof sketch). *Under the trace constraint $\mathrm{tr}(\Sigma_B) \approx B$, the term $\mathrm{tr}(\Sigma_B) + B$ in equation 22 is effectively constant, so minimizing $d_B^2(\Sigma_B, I_B)$ is equivalent to maximizing $\mathrm{tr}(\Sigma_B^{1/2})$, see equation 23. Equation equation 25 shows that $\mathrm{tr}(\Sigma_B^{1/2})$ equals the nuclear norm of $Z$. Thus any $Z$ that maximizes $\|Z\|_*$ (under the same trace constraint) also minimizes $d_B^2(\Sigma_B, I_B)$, and vice versa.*

**On the 'Mild' Trace Constraint and Approximation Quality.** The "mild" nature of our trace constraint stems from its satisfaction by standard normalization practices in modern deep learning. We distinguish between two common scenarios.

**Exact Equivalence with $\ell_2$ Normalization.** Many GCD and representation learning methods apply $\ell_2$ normalization to the final embeddings $z_i$. If each row $z_i$ of the stacked matrix $Z$ is $\ell_2$-normalized such that $\|z_i\|_2 = 1$, then the trace of the Gram matrix $\Sigma_B = ZZ^\top$ becomes a strict constant:

$$\mathrm{tr}(\Sigma_B) = \mathrm{tr}(ZZ^\top) = \sum_{i=1}^{B} \|z_i\|_2^2 = \sum_{i=1}^{B} 1 = B.$$

In this scenario, the $\mathrm{tr}(\Sigma_B)$ term in the Bures distance formula is constant, and minimizing the Bures distance becomes *exactly equivalent* to maximizing the nuclear norm $\|Z\|_*$, as shown in equation 23 and equation 25.

**High-Fidelity Approximation with LayerNorm.** Our method is applied after a LayerNorm layer, a standard component in Transformer architectures. While LayerNorm does not strictly enforce $\|z_i\|_2 = 1$, it normalizes the features of each sample to have zero mean and unit variance across the feature dimension, followed by a learned affine transformation. This operation ensures that the row norms $\|z_i\|_2$ are tightly concentrated around a stable value during training, making $\mathrm{tr}(\Sigma_B)$ nearly constant. This renders the nuclear norm an extremely high-fidelity and empirically effective surrogate for the Bures distance objective. Our experiments confirm this: using the exact Bures loss versus the nuclear-norm surrogate yields nearly identical training dynamics and final performance on all GCD benchmarks, validating that Lemma 3 captures the relevant regime for our method.

# D    MORE ANALYSIS

## D.1    COMPARISON WITH SELF-SUPERVISED LEARNING SCHEMES

In Section 5.4 we empirically compare BIA with two representative isotropy-related regularizers, VICReg (Bardes et al., 2022) and CorInfoMax (Ozsoy et al., 2022). Here we give a concise but more detailed analysis of how these objectives differ and why BIA is particularly effective in the GCD setting.

### D.1.1    OBJECTIVES AND LEVEL OF OPERATION

**VICReg.** VICReg is a self-supervised pre-training method combining three terms: (i) an invariance loss between two augmented views, (ii) a per-dimension variance term enforcing non-degenerate variance, and (iii) a covariance term penalizing off-diagonal entries of the feature covariance. Given a batch of features $Z \in \mathbb{R}^{B \times d}$, the covariance penalty acts on $\mathrm{Cov}(Z)$ via

$$L_{\mathrm{cov}} = \sum_{i \neq j} \mathrm{Cov}(Z)_{ij}^2,$$

while the variance term keeps each $\mathrm{Var}(Z)_i$ above a threshold. Thus VICReg regularizes features at the *instance level* and per-coordinate statistics.

**CorInfoMax.** CorInfoMax is also proposed in a self-supervised context and aims to maximize mutual information between representations and an auxiliary target distribution. It discourages trivial collapse by requiring representations to remain informative, but does not explicitly control the eigen-spectrum of the covariance matrix.

**BIA.** BIA instead operates on the *batch class-token matrix* used directly for GCD decisions. Let $Z \in \mathbb{R}^{B \times d}$ be the stacked class tokens after LayerNorm, and let $\Sigma_B = ZZ^\top \in \mathbb{R}^{B \times B}$ denote the batch Gram. Since LayerNorm makes each row satisfy $\|z_i\|_2 \approx 1$, we have

$$\mathrm{tr}(\Sigma_B) = \sum_{i=1}^{B} \|z_i\|_2^2 \approx B,$$

which is nearly constant. BIA minimizes the Bures distance between $\Sigma_B$ and $I$,

$$d_B^2(\Sigma_B, I) = \text{tr}(\Sigma_B) + B - 2\,\text{tr}(\Sigma_B^{1/2}).$$

Under the trace constraint this is equivalent to *maximizing* $\text{tr}(\Sigma_B^{1/2})$. If $\{\mu_j\}$ are the eigenvalues of $\Sigma_B$, then

$$\text{tr}(\Sigma_B^{1/2}) = \sum_j \sqrt{\mu_j} = \sum_j s_j(Z) = \|Z\|_*,$$

the nuclear norm of $Z$. Hence BIA can be seen as a spectrum-shaping objective that maximizes a concave function of the eigenvalues of $\Sigma_B$ under a mild trace constraint.

In summary, VICReg and CorInfoMax act primarily at the instance/coordinate level, whereas BIA directly regularizes the batch class-token Gram in the space where GCD methods perform classification, clustering, and prototype updates.

### D.1.2 Instance-wise vs. Spectrum-wise Regularization

A key distinction is that VICReg and CorInfoMax are essentially *coordinate-wise* or *instance-wise* regularizers, while BIA is explicitly *spectrum-wise*.

The covariance term of VICReg drives $\text{Cov}(Z)$ towards a diagonal matrix with controlled diagonal entries. This decorrelates coordinates and constrains per-dimension variance, but does not directly reason about the global shape of the eigen-spectrum beyond these coordinate-level constraints. CorInfoMax promotes informative representations, but information can still concentrate in a low-dimensional subspace; there is no explicit mechanism to prevent highly unbalanced eigenvalues.

By contrast, BIA treats the eigenvalues $\{\mu_j\}$ of $\Sigma_B$ as a whole. Under $\sum_j \mu_j = \text{tr}(\Sigma_B) \approx B$, maximizing $\sum_j \sqrt{\mu_j}$ is a Schur-concave objective: it favors eigenvalue configurations that are more uniform. Intuitively, BIA redistributes energy from overly dominant principal components to smaller ones, increasing the effective rank and von Neumann entropy of the class-token autocorrelation matrix. This directly targets the dimensional collapse and skewed energy distribution observed in GCD.

### D.1.3 Alignment with the GCD Setting

The GCD setting introduces two challenges that are absent in standard self-supervised pre-training:

- unlabeled batches contain a mixture of known and novel classes; and
- pseudo-labels for novel classes are noisy, often combined with class imbalance and domain gap.

In this regime, the three regularizers behave differently:

**VICReg.** The invariance term encourages strong instance-level invariance. With noisy pseudo-labels and fine-grained novel categories, this can over-compress intra-class variability and merge distinct novel sub-classes into overly compact clusters. The coordinate-wise covariance penalty may also suppress directions that remain informative for subtle novel distinctions, making performance sensitive to the loss weight and augmentation strength.

**CorInfoMax.** Maximizing mutual information can encode both signal and noise. Under noisy pseudo-labels, high mutual information with an imperfect target does not guarantee a well-conditioned geometry and can reinforce spurious correlations. Without an explicit anti-collapse or spectrum-flattening term, CorInfoMax does not systematically correct the eigen-structure drift induced by GCD training.

**BIA.** BIA acts only on the geometry induced by the underlying GCD loss (e.g., SimGCD, CMS). The GCD objective shapes semantic directions by pulling samples of the same (pseudo-)class together and pushing different classes apart. BIA does not attempt to re-learn this structure; instead, it prevents it from collapsing into a few dominant directions. By increasing the effective rank and entropy of

the class-token Gram, BIA enlarges the manifold capacity available to encode intra-class variability, which is crucial for distinguishing multiple novel categories that differ only subtly. Since BIA is defined at the batch class-token level, where prototypes and the number of clusters are estimated, a better-conditioned Gram leads to more stable prototype updates and sharper decision boundaries under noisy pseudo-labels.

Empirically, this alignment with the GCD decision space translates into consistent gains in overall and, in particular, novel-class accuracy when BIA is plugged into strong GCD baselines, whereas directly transplanting VICReg or CorInfoMax often yields mixed or fragile improvements.

### D.1.4 ROBUSTNESS AND HYPERPARAMETER SENSITIVITY

Finally, we observe a practical difference in robustness. VICReg and CorInfoMax introduce strong instance-level or information-theoretic constraints whose interaction with pseudo-label noise and class imbalance is highly sensitive to loss weights and augmentation policies; hyperparameters tuned on one backbone or dataset do not transfer easily.

BIA instead relies on a single scalar weight $\lambda$ and the mild trace constraint from LayerNorm. Because it only reshapes the spectrum of already task-aligned class tokens, its effect is more uniform across backbones and datasets. Our ablations show that BIA is stable over a wide range of $\lambda$ and feature dimensionality, and we use essentially the same $\lambda$ across all GCD methods and benchmarks.

**Discussion.** Overall, VICReg and CorInfoMax are strong self-supervised methods, but are not tailored to the mixed known/novel, pseudo-label-driven nature of GCD. BIA is explicitly designed as a batch class-token spectrum regularizer under a trace constraint, complementary to existing GCD objectives. This design explains why BIA yields more consistent gains on both overall and novel-category accuracy, as well as improved class-number estimation, in our experiments.

### D.1.5 DETAILED COMPARISON AND HYPERPARAMETER ANALYSIS OF SSL METHODS

To provide a more thorough assessment of BIA against other isotropy-promoting schemes, we offer a detailed analysis of the compared self-supervised learning (SSL) methods, CorInfoMax and VICReg. Our goal is to clarify their design motivations and conduct a new, in-depth hyperparameter analysis to investigate their sensitivity within the GCD context.

**Design Motivations and Implementation Details.** First, we clarify the design and implementation of the compared methods.

- **CorInfoMax** is an SSL method that maximizes the mutual information between representations. Its loss function consists of a similarity term to align different views of the same sample and a covariance term to regularize the feature covariance matrix. Following the original paper, we set the internal weights for the similarity loss to 500 and the covariance loss to 1. For the overall loss coefficient $\lambda$, we searched around the value used for BIA (0.004) and found this to be optimal for our main experiments.

- **VICReg** is an SSL method that learns representations by enforcing three principles: an **invariance** term (aligning augmented views), a **variance** term (preventing informational collapse along feature dimensions), and a **covariance** term (decorrelating feature dimensions). The full loss is a weighted sum of these three components.

**Hyperparameter Sensitivity in the GCD Context.** We hypothesize that for the GCD task, VICReg's **invariance** term is conceptually redundant with the contrastive objectives already present in GCD baselines like SimGCD. Therefore, the core of its isotropy-promoting effect resides in the **variance** and **covariance** penalties, controlled by coefficients $\mu$ and $\nu$, respectively.

To investigate this and ensure a fair comparison, we conducted a new, comprehensive hyperparameter sweep. We integrated only VICReg's variance and covariance terms into the SimGCD baseline and varied their respective coefficients on three fine-grained datasets. The results are summarized in Table A1, Table A2, and Table A3.

Table A1: Hyperparameter ablation for VICReg's uniformity loss components (variance coeff. $\mu$ and covariance coeff. $\nu$) on the **CUB** dataset, integrated into SimGCD. We report All/Old/New accuracy (%).

| $\nu \setminus \mu$ | 10 | 25 | 40 |
|---|---|---|---|
| 0.2 | 57.3 / 59.9 / 55.4 | 58.3 / 62.7 / 56.0 | 60.2 / 64.5 / 57.6 |
| 1 | 56.5 / 61.2 / 53.9 | 61.1 / 66.0 / 58.1 | 59.8 / 62.1 / 58.5 |
| 5 | **61.3** / 62.7 / 60.0 | 60.1 / 66.5 / 57.2 | 59.5 / 64.3 / 57.1 |

Table A2: Hyperparameter ablation for VICReg's uniformity loss components on the **Stanford Cars** dataset.

| $\nu \setminus \mu$ | 10 | 25 | 40 |
|---|---|---|---|
| 0.2 | 50.6 / 67.8 / 42.3 | 51.1 / 66.5 / 43.4 | 50.6 / 65.9 / 43.7 |
| 1 | 51.1 / 67.7 / 42.3 | 52.0 / 68.6 / 44.1 | 47.1 / 62.7 / 38.7 |
| 5 | **52.1** / 69.1 / 42.1 | 49.3 / 65.4 / 42.5 | 48.2 / 66.9 / 39.8 |

Table A3: Hyperparameter ablation for VICReg's uniformity loss components on the **FGVC Aircraft** dataset.

| $\nu \setminus \mu$ | 10 | 25 | 40 |
|---|---|---|---|
| 0.2 | 52.5 / 56.1 / 50.8 | 52.4 / 60.0 / 48.9 | 52.6 / 55.6 / 50.8 |
| 1 | 52.2 / 56.2 / 50.1 | 54.6 / 56.2 / 53.8 | **54.8** / 55.6 / 53.5 |
| 5 | 48.8 / 54.7 / 45.7 | 49.3 / 59.1 / 44.7 | 50.7 / 56.2 / 47.8 |

**Conclusion from SSL Methods.** Our new experiments reveal two important findings. First, the performance of VICReg is highly sensitive to the choice of hyperparameters, and the optimal setting varies significantly across datasets. For instance, on CUB (Table A1), the best All accuracy (61.3%) is achieved with $(\mu, \nu) = (10, 5)$, while on Stanford Cars (Table A2), the best result (52.1%) requires $(\mu, \nu) = (10, 5)$. Second, even after this extensive tuning, the best-performing VICReg configuration on each dataset still does not surpass the results of our proposed BIA method (BIA achieves 62.1% All accuracy on SimGCD+CUB, outperforming most VICReg settings and being more stable than its peak). BIA remains robust and uses a single hyperparameter setting across all datasets.

These findings reinforce our conclusion: while general isotropy regularizers like VICReg can be beneficial, BIA provides a more direct, robust, and effective solution specifically tailored to the geometric challenges of GCD.

## D.2 COMPARISON WITH ISOTROPY-ENCOURAGING REGULARIZERS

### D.2.1 INTRODUCTION OF ISO-FROB AND ISO-ENT

In addition to VICReg and CorInfoMax, we also considered two simple, more "direct" isotropy objectives applied to the batch class-token Gram matrix $\Sigma_B = ZZ^\top$:

$$L_{\text{iso-frob}} = \left\| \Sigma_B - \frac{\text{tr}(\Sigma_B)}{B} I \right\|_F^2, \tag{26}$$

$$L_{\text{iso-ent}} = -H(\tilde{\Sigma}_B), \quad \tilde{\Sigma}_B = \frac{\Sigma_B}{\text{tr}(\Sigma_B)}, \tag{27}$$

where $H(\tilde{\Sigma}_B) = -\text{tr}(\tilde{\Sigma}_B \log \tilde{\Sigma}_B)$ is the von Neumann entropy. We refer to these as *Iso-Frob* and *Iso-Ent*, respectively. All three objectives (BIA, Iso-Frob, Iso-Ent) promote more isotropic geometry, but they differ in how they act on the eigenvalues of $\Sigma_B$ and how strongly they constrain the covariance, which has practical consequences for GCD.

**Spectral view under a trace constraint.** Let $\{\mu_j\}_{j=1}^B$ be the eigenvalues of $\Sigma_B$. Because we apply BIA after LayerNorm on class tokens, each row $z_i$ satisfies $\|z_i\|_2 \approx 1$ and thus $\text{tr}(\Sigma_B) = \sum_j \mu_j \approx B$

is nearly constant. In this regime, all three losses can be interpreted as functions on the simplex

$$\mathcal{S}_B = \big\{ \mu \in \mathbb{R}_+^B : \sum_{j=1}^B \mu_j = B \big\}.$$

**BIA.** The Bures loss is

$$d_B^2(\Sigma_B, I) = \text{tr}(\Sigma_B) + B - 2\,\text{tr}(\Sigma_B^{1/2}), \tag{28}$$

which, up to an additive constant, is equivalent to maximizing $\text{tr}(\Sigma_B^{1/2}) = \sum_j \sqrt{\mu_j}$ under $\sum_j \mu_j = B$. Hence the effective spectral objective is

$$f_{\text{BIA}}(\mu) = -\sum_{j=1}^B \sqrt{\mu_j}. \tag{29}$$

The map $\mu \mapsto \sum_j \sqrt{\mu_j}$ is symmetric and concave on $\mathcal{S}_B$, i.e. Schur-concave; it is maximized at the uniform point ($\mu_j = 1$) and prefers more balanced spectra, but does not force any eigenvalue to match a fixed target. In particular, small eigenvalues receive relatively large positive gradients ($\partial \sqrt{\mu_j}/\partial \mu_j = \frac{1}{2}\mu_j^{-1/2}$), which gently lifts collapsed directions without overly penalizing moderate anisotropy.

**Iso-Frob.** For Iso-Frob, the spectral form of equation 26 under $\sum_j \mu_j = B$ is

$$f_{\text{Frob}}(\mu) = \sum_{j=1}^B \big(\mu_j - 1\big)^2. \tag{30}$$

This objective also has its minimum at the uniform point, but it penalizes *quadratic* deviations from the exact spherical target $\mu_j = 1$ along each eigen-direction. As a result, Iso-Frob behaves like a strict whitening penalty: any structured anisotropy (even if it is semantically meaningful for separating classes) is penalized proportionally to the squared deviation. In the GCD setting, where batches mix known and novel classes and pseudo-labels are noisy, we find this rigidity undesirable:

- When pseudo-labels are imperfect, some anisotropy reflects meaningful semantic structure; aggressively driving $\Sigma_B$ towards a scaled identity can partially undo class separation learned by the GCD loss.
- Quadratic penalties yield gradients that grow linearly with $|\mu_j - 1|$, so large deviations (e.g. caused by outliers or class imbalance) dominate the update and can lead to over-regularization of a few directions.

Empirically, Iso-Frob improves baselines modestly but is consistently weaker and less stable than BIA in our GCD experiments.

**Iso-Ent.** For Iso-Ent, using the normalized spectrum $\tilde{\mu}_j = \mu_j / \sum_k \mu_k = \mu_j/B$, the von Neumann entropy reduces to Shannon entropy:

$$H(\tilde{\Sigma}_B) = -\sum_{j=1}^B \tilde{\mu}_j \log \tilde{\mu}_j, \quad f_{\text{Ent}}(\mu) = -H(\tilde{\Sigma}_B) = \sum_{j=1}^B \tilde{\mu}_j \log \tilde{\mu}_j. \tag{31}$$

This is again a symmetric, Schur-concave function maximized at the uniform spectrum. However, its gradient with respect to $\mu_j$ is proportional to $\log \tilde{\mu}_j + 1$, which diverges as $\tilde{\mu}_j \to 0$. Thus Iso-Ent strongly amplifies very small eigenvalues, making it numerically and statistically sensitive:

- Small eigenvalues, which may correspond to noise or spurious directions induced by incorrect pseudo-labels, receive extremely large updates and can be over-emphasized.
- Computing matrix logarithms and the corresponding gradients is less stable than the SVD-based square root used in BIA, especially when $\Sigma_B$ is near-singular early in training.

### D.2.2 COMPARISON WITH ISOTROPY REGULARIZERS UNDER VARYING BATCH SIZES

To further situate BIA, we conducted additional experiments comparing it against two simple, more "direct" isotropy objectives applied to the class-token Gram matrix $\Sigma_B$. This analysis examines the

Table A4: Comparison with more isotropy-encouraging regularizers on the CUB dataset.

| Regularizer + Batch Size | CUB (All) | CUB (Known) | CUB (Novel) |
|---|---|---|---|
| **Iso-Frob** | | | |
| 32 | 51.9 | 55.9 | 49.9 |
| 64 | 60.3 | 70.7 | 55.3 |
| 128 | 61.5 | 69.4 | 57.4 |
| **Iso-Ent** | | | |
| 32 | 48.8 | 56.0 | 45.2 |
| 64 | 59.3 | 68.9 | 54.9 |
| 128 | 61.8 | 68.1 | 57.1 |
| **BIA (Ours)** | | | |
| 32 | 53.1 | 58.5 | 50.4 |
| 64 | 59.5 | 69.3 | 54.6 |
| 128 | 62.1 | 65.8 | 60.3 |

robustness of each regularizer to changes in batch size, a critical factor for training stability and performance.

We integrated these regularizers and BIA into the SimGCD baseline and evaluated their performance on three fine-grained datasets with batch sizes of 32, 64, and 128. The results are presented in Table A4.

As shown in Table A4, our analysis yields several key insights. First, BIA consistently outperforms both Iso-Frob and Iso-Ent across all datasets and batch sizes, demonstrating its superior effectiveness. Second, BIA exhibits remarkable robustness to variations in batch size, with performance remaining stable from 32 to 128. In contrast, both Iso-Frob and, particularly, Iso-Ent show greater sensitivity, with performance degrading more noticeably at smaller batch sizes.

These empirical results align with our theoretical understanding. (i) **Iso-Frob** enforces a rigid whitening penalty, attempting to match an exactly spherical covariance. This can be overly restrictive in the noisy GCD setting, where some degree of structured anisotropy learned from pseudo-labels may be beneficial for class separation. The aggressive regularization can thus suppress useful semantic information. (ii) **Iso-Ent**, which involves matrix logarithms, is known to be numerically sensitive, especially to small eigenvalues that are common early in training or with small batches. This sensitivity likely contributes to its more volatile performance.

In contrast, BIA's Bures / nuclear-norm formulation avoids these pitfalls. It maximizes a concave function of the eigenvalues under a mild trace constraint, which encourages a more uniform spectrum by *reshaping* it (lifting smaller eigenvalues and gently suppressing larger ones) without forcing all directions to a specific target. This more flexible and robust mechanism for promoting isotropy appears exceptionally well-suited to the noisy, mixed known/novel regime of GCD.

### D.3 IMPACT OF EMBEDDING QUALITY

In Table 2, the accuracy gains on the CIFAR100 and Herbarium19 datasets are marginal. We use this as a starting point to analyze the conflict between enhancing feature completeness and low embedding quality in GCD. DINO, through self-supervision, already has a good feature representation capability, but due to the distribution of data, its embedding quality remains low. One source of low quality is the data size, and the other is data semantics.

(1) Specifically, when the small-sized CIFAR100 images are interpolated and input into ViT, the high-frequency information is lost. For example, when identifying animal categories, the low-frequency features such as the outline of the animal may be captured relatively well, but the detailed features such as the texture and eyes of the animal (high-frequency features) are difficult to accurately extract. In this case, the model can only cluster through some shortcut information, rather than accurately clustering based on the complete intra-class features. Since the manifold dimension of the low-frequency features is relatively low, it is unable to fully capture the diversity and complexity within

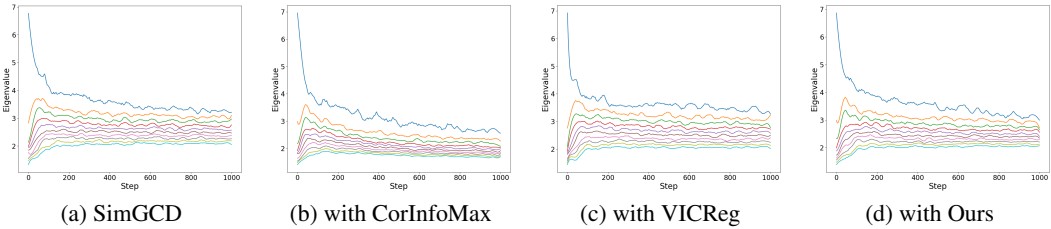

| (a) SimGCD | (b) with CorInfoMax | (c) with VICReg | (d) with Ours |

Figure A1: Trends in the top 10 eigenvalues as the number of training steps grows.

the class. Therefore, enhancing the completeness of the intra-class representation on small-sized data is challenging.

(2) Herbarium19 is a large-scale herbal plant recognition dataset, which is not in the model's training data and inherently cannot provide highly discriminative representations. Additionally, the large number of categories makes the decision boundary more chaotic, and existing GCD schemes cannot cluster well. Therefore, enhancing the completeness of intra-class representation on overly low-quality embeddings is not feasible, as the overlap of feature spaces across categories is too large, and samples within a cluster come from multiple categories.

### D.4 Analysis on the Evolution of Eigenvalues

Compared to the two optimization directions, VICReg and CorInfoMax, BIA offers a smoother and more uniform convergence of feature values, addressing some key limitations in both methods (Figure A1). VICReg, as a variance-based regularization approach, promotes feature variance and decorrelation but lacks explicit emphasis on intra-class representation completeness. This results in less expressive class boundaries and less effective fine-grained category separation. CorInfoMax, on the other hand, focuses on maximizing mutual information between features and their target distribution but does not sufficiently prevent dimensional collapse or guarantee richer intra-class representations. Both methods, while effective in some contexts, fail to fully capture the complex, high-dimensional structure of the data.

In contrast, BIA directly targets the manifold capacity of class tokens, ensuring that intra-class representations remain complete and informative. By maximizing the nuclear norm of the class token's singular values, BIA ensures that feature values converge uniformly, without the collapse seen in other methods. This leads to more robust and accurate clustering, particularly when discovering novel categories. The smooth convergence of BIA reflects its ability to optimize representation quality while maintaining high inter-class separability, which is critical for open-world learning tasks.

### D.4.1 Computational Overhead Analysis

We quantitatively analyze the computational overhead of BIA to substantiate our claim that it is a lightweight, plug-and-play module.

**Empirical Validation.** To verify this empirically, we conducted two sets of measurements. First, we measured the wall-clock time of the SVD step (the most complex part of BIA) and its relative overhead compared to the backbone's computation time. As shown in Table A5, the SVD step's contribution is minimal, accounting for less than 1.5% of the backbone's compute time even with a batch size of 256. This confirms that the core operation of BIA is negligible in practice.

Table A5: Relative wall-clock time overhead of the SVD computation in BIA with respect to the ViT backbone's forward-backward pass, measured across different batch sizes.

| Batch Size | 64 | 128 | 192 | 256 |
|---|---|---|---|---|
| **Time Overhead (%)** | 0.37% | 0.81% | 1.01% | 1.47% |

Second, we evaluated the total wall-clock time per epoch and peak GPU memory for representative GCD frameworks with and without BIA. The results in Table A6 show that integrating BIA increases

the total training time by less than 1% and the peak memory usage by only a few tens of megabytes across diverse methods like SelEx, SimGCD, CMS, and SPTNet.

Table A6: Total training time per epoch and peak GPU memory overhead of BIA when integrated into representative GCD frameworks on the CUB dataset.

| Method | SelEx | SimGCD | CMS | SPTNet |
|---|---|---|---|---|
| *Peak GPU Memory (MB)* | | | | |
| Original | 8640 | 6354 | 6040 | 23322 |
| **+ BIA (Ours)** | 8674 | 6390 | 6074 | 23670 |
| *Time per Epoch (s)* | | | | |
| Original | 24.99 | 24.50 | 28.81 | 22.09 |
| **+ BIA (Ours)** | 25.16 | 24.67 | 28.99 | 22.26 |

Collectively, these results provide strong quantitative evidence that BIA is computationally efficient, imposing a negligible burden on standard training pipelines while delivering significant performance gains.

