# OpenReview forum: "Bures-Isotropy Alignment: Manifold Learning of Generalized Category Discovery"
_ICLR.cc/2026/Conference — ICLR 2026 Poster_

### Official Review · Reviewer_trb7 · 2025-10-30

**Soundness:** 3
**Presentation:** 3
**Contribution:** 2
**Rating:** 6
**Confidence:** 1

**Summary:**

This paper addresses a core challenge in the Generalized Category Discovery (GCD) task: characterizing geometric quality degradation. It proposes an innovative method called Bures-Isotropy Alignment. GCD aims to simultaneously identify known categories and discover novel categories from unlabeled data.

**Strengths:**

Instead of following mainstream approaches to further pursue feature compression, this paper astutely points out the fundamental bottleneck in current GCD research: the degradation of representational geometric quality. It offers a novel and insightful solution from the perspective of "restoring geometry" rather than "compressing features."

The paper successfully introduces the Bures metric from quantum information science into the field of machine learning and cleverly derives that minimizing it is equivalent to maximizing the nuclear norm of the feature matrix.

**Weaknesses:**

1. The paper emphasizes its connection to the literature on representation collapse, but could delve deeper into the specific differences and connections between BIA and other methods in self-supervised learning aimed at solving similar problems, thus more clearly defining its unique contribution.

2. The paper's core argument is "geometry recovery," but BIA is essentially still an optimization objective that indirectly "recovers" geometry by changing the optimization direction. A more in-depth discussion could explore how this optimization dynamically corrects the geometric structure during training, rather than just post-hoc statistical analysis.

3. The paper mentions that BIA has low computational overhead, but provides no quantitative data (such as the percentage increase in training time or changes in GPU memory usage) to support this claim.

**Questions:**

See Weaknesses

---

> ### Author Response · Authors · 2025-11-22
> **Responses to Reviewer trb7 (Part 1)**
>
> We appreciate the reviewer for considering our work pointing out fundamental bottleneck and offering a novel and insightful solution. Our detailed responses to your comments are as follows.
>
>
>
>
> > W1: Differences and connections between BIA and other methods in self-supervised learning.
>
> We thank the reviewer for this insightful comment. We agree that the paper can better describe both the connections and the key distinctions between BIA and existing self-supervised anti-collapse methods. Below, we clarify these relations and highlight BIA’s unique contribution to GCD.
>
> **1. Connections to self-supervised anti-collapse methods**: Our work connects to methods in self-supervised learning (SSL) that mitigate feature collapse:
>
> *   **VICReg** promotes non-collapsed representations by regularizing the feature covariance matrix. It encourages coordinate-wise decorrelation through its covariance loss, $L_{\text{cov}} = \sum_{i\neq j} \mathrm{Cov}(\boldsymbol{Z})_{ij}^2$, while also maintaining variance along each feature dimension.
>
> *   **CorInfoMax** avoids collapse by maximizing the mutual information between representations, which implicitly encourages the features to remain informative and spread out.
>
> These approaches share our high-level goal of promoting a well-conditioned feature space and preventing representational degeneracy by effectively operating at the instance or coordinate level.
>
> **2. Key differences and BIA's unique contribution for GCD**: The primary distinction of BIA lies in its targeted design for the unique challenges of GCD, for which generic SSL regularizers are not optimized. The GCD setting relies on clustering unlabeled data containing mixed known/novel classes, often with noisy pseudo-labels.
>
> BIA's unique contribution is performing **spectrum-wise regularization directly within the GCD decision space**.
> *   **Level of operation.** Instead of regularizing per-coordinate statistics of instance features, BIA operates directly on the batch class-token Gram matrix, $\Sigma_B = \boldsymbol{Z}\boldsymbol{Z}^\top$, where $\boldsymbol{Z} \in \mathbb{R}^{B\times d}$ is the matrix of stacked class tokens used for clustering. This is the space where GCD methods make decisions, update prototypes, and estimate the number of classes.
> *   **Mechanism.** By maximizing the nuclear norm $\|\boldsymbol{Z}\|_*$, BIA directly reshapes the *entire eigenvalue spectrum* of the Gram matrix. This process explicitly redistributes spectral energy from a few dominant directions to weaker ones, increasing the effective rank and von Neumann entropy of the class-token geometry. This directly counteracts the dimensional collapse we observe in GCD training.
> *   **Suitability for GCD.** This geometry-aware approach is fundamentally different from the instance-level constraints of VICReg or the information-theoretic objective of CorInfoMax. It is specifically designed to stabilize the feature manifold against the noise and class imbalance inherent to GCD, which is a problem that coordinate-wise decorrelation or generic information maximization do not directly address.
>
> To summarize, while other methods prevent collapse in a general sense, BIA is the first method to perform targeted geometric restoration on the class-token manifold, aligning the optimization objective with the decision-making process of GCD.
>
> We incorporated a concise version of this comparison into Section 5.4 and expand upon this detailed analysis in Appendix C.1 to better highlight BIA's novelty and specific contribution to the GCD task.

---

> ### Author Response · Authors · 2025-11-22
> **Responses to Reviewer trb7 (Part 2)**
>
> > W2: How does this optimization dynamically correct the geometric structure during training?
>
> We fully agree that understanding how BIA dynamically corrects the geometry during training is essential.  In fact, this is exactly the purpose of Figure 7 in the appendix, where we track how the top-10 eigenvalues of the class-token Gram evolve as training progresses for four variants: SimGCD, SimGCD+CorInfoMax, SimGCD+VICReg, and SimGCD+BIA.
>
> Below we summarize what Figure 7 reveals about the *geometric structure* and how BIA corrects it over time:
>
> * **Baseline SimGCD.** For SimGCD alone, the top-10 singular values grow and remain *well separated* throughout training. A few leading directions dominate, and the gaps between adjacent singular values stay large. Geometrically, this indicates that the class-token manifold is increasingly squeezed into a low-dimensional subspace, the representation capacity gets concentrated in a handful of principal components, while the remaining directions are under-utilized. This is precisely the kind of anisotropic, low-rank geometry that harms discovery of many fine-grained novel classes.
>
> * **With CorInfoMax.** Adding CorInfoMax changes the overall scale but keeps the *imbalance* between singular values. The top-1 component remains extremely large and clearly detached from the rest across the whole training trajectory. This indicates that although mutual-information maximization keeps the representation “informative”, it still allows a single dominant direction to capture most of the energy, so the global geometry remains collapsed and highly anisotropic.
>
> * **With VICReg.** With VICReg, the top-10 singular values become more *spread out in magnitude*, but they fluctuate and do not gradually move toward a compact, balanced spectrum. This reflects the coordinate-wise nature of VICReg that it increases variance and decorrelation per feature dimension, but does not directly organize the eigenvalue distribution of the class-token Gram. As a result, the manifold is “noisy and scattered” rather than forming a well-conditioned, high-capacity structure tailored to GCD.
>
> * **With BIA (ours).** In contrast, with BIA we observe a *two-phase dynamic correction* of geometry. At early steps, the top-1 singular value becomes relatively large, allowing the model to quickly explore the main semantic direction and form coarse cluster structure. As training continues, BIA’s nuclear-norm/Bures objective gradually **pulls the remaining singular values up and closes the gaps** between them so the top-10 values become more uniform and closer to each other. Geometrically, this means that the class-token manifold evolves from a highly skewed shape to a well-conditioned, high-rank, nearly isotropic structure, where semantic information is distributed across multiple principal components instead of collapsing into a few.
>
> This temporal behavior is exactly what we mean by *dynamic geometry recovery*. BIA does not merely report a high von Neumann entropy after training. Instead, it actively reshapes the spectrum of the class-token Gram at every step, preventing the GCD objective from collapsing the manifold and maintaining sufficient capacity to represent subtle novel categories. To make this clearer, we added a concise summary of this spectral evolution in Section 5.4, ensuring the connection between optimization dynamics and geometric structure clearer.

---

> ### Author Response · Authors · 2025-11-22
> **Responses to Reviewer trb7 (Part 3)**
>
> > W3: Computational overhead.
>
> To reassure the reviewer, we measure the increased time overhead as we vary the batch size in Table R1.  From a complexity perspective, the additional computation introduced by BIA depends only on the batch size $B$. Per batch, BIA adds:
>
> * a batch Gram computation $\Sigma_B = ZZ^\top$ with cost $O(B^2 d)$;
> * an eigendecomposition/SVD of a $B \times B$ matrix with cost $O(B^3)$.
>
> In contrast, the ViT backbone forward–backward pass scales with both $B$ and the number of tokens and dominates the runtime. In standard GCD settings we use $B=128$, so the added cost is theoretically small.
>
> To substantiate this, we measure the wall-clock time of the SVD step for different batch sizes.
>
> Table R1: Relative time overhead of the SVD step.
>
> |Batch Size|64|128|192|256|
> |-|-|-|-|-|
> |Time(s)|1.39e-05(0.37%)|2.94e-05(0.81%)|3.79e-05(1.01%)|5.42e-05(1.47%)|
>
> Across $B = 64, 128, 192, 256$, the SVD contributes **below 1.5%** of the backbone compute per training step, confirming that its cost is negligible in practice.
>
> We further evaluate wall-clock time per epoch and peak GPU memory for a representative configuration SelEx & CMS (contrastive learning), SimGCD (prototype learning) and SPTNet (prompt learning) in Table R2.
>
> Table R2: Training time and memory overhead of BIA on CUB.
>
> |Costs|SelEx|SimGCD|CMS|SPTNet|
> |-|-|-|-|-|
> |GPU memory (MB)| | | | |
> |Original|8640|6354|6040|23322|
> |**Ours**|8674|6390|6074|23670|
> |Time (s)| | | | |
> |Original|24.99|24.50|28.81|22.09|
> |**Ours**|25.16|24.67|28.99|22.26|
>
> The results show that BIA increases training time by less than 1% and peak memory by only a few tens of MB under standard GCD configurations. We include these tables in the Appendix C.5, so that our “low computational overhead” claim is quantitatively supported.
>
> ---
>
> We thank the reviewer for the thoughtful and constructive critiques. If there are any further questions or reflections, we are eager to continue this meaningful discussion.

---

### Official Review · Reviewer_aYVi · 2025-10-30

**Soundness:** 4
**Presentation:** 3
**Contribution:** 3
**Rating:** 8
**Confidence:** 4

**Summary:**

This paper proposes a novel loss function, termed Bures-Isotropy Alignment (BIA) loss, which aims to enhance intra-class representation completeness, while maintaining inter-class separability in the context of Generalized Category Discovery (GCD)—a task involving clustering over both known and unknown categories. The BIA loss is inspired by the insight that when class-token covariance matrices, constructed from class-specific representations, approach an isotropic form, the learned representation space becomes more completed and less susceptible to collapse. Mathematically, the loss is formulated as the nuclear norm of stacked class representations based on the Bures-Isotropy alignment principle. Experimental results show that incorporating the BIA loss into baseline GCD models, *i.e.*, SelEX, SimGCD, CMS, SPTNet, consistently improves accuracy across various datasets and architectures, for both known and unknown categories. Furthermore, the method achieves superior performance in category number estimation, reinforcing its overall effectiveness.

**Strengths:**

- The motivation and insight that encouraging class-token covariance to become isotropic leads to more robust within-class representations is both intuitive and compelling. The theoretical support using Bures-Isotropy alignment (Section 3.3) is mathematically well-grounded and strengthens the conceptual foundation.

- The process of deriving a simple and practical loss formulation from this motivation is clear and elegant, making the approach easily applicable to future work in the GCD research community.

- The experimental design is thorough and effectively demonstrates both the simplicity and generality of the proposed method.

- Incorporating the BIA loss consistently leads to improved performance in Generalized Category Discovery tasks and category number estimation across various datasets and baseline models, highlighting the robustness and versatility of the approach.

**Weaknesses:**

I do not have any major concerns with this submission; however, I have several minor suggestions for clarification and completeness.

- In Section 3.4, the paper shows that incorporating the BIA loss during training improves Neumann Entropy, which measures the uniformity of class-specific representation distributions. However, since both BIA and Neumann Entropy are defined in a similar manner—being proportional to the eigenvalues—the observed positive correlation between the two is somewhat expected. It would be helpful to clarify what specific insight or takeaway this section intends to convey beyond confirming this relationship.

- Section 5.4 presents interesting observations, but without sufficient details about the compared methods (e.g., CorInfoMax and VICReg), it is difficult to fully understand or assess the comparison. Please provide additional details or the design motivations of these methods to improve clarity.

- Figure 1 is currently not referenced anywhere in the manuscript. It should be mentioned and discussed in an appropriate place to ensure proper integration into the paper’s narrative.

**Questions:**

**Is the proposed BIA loss differentiable?**

In other words, is the process of computing eigenvalues from the covariance matrix differentiable? It appears that it should be, since (i) both matrix decomposition and squared eigenvalue operations, involved in loss calculation, have known derivative forms, and (ii) the paper reports successful training results using this loss.

I would appreciate clarification on whether the authors have a direct explanation or confirmation regarding the differentiability of the BIA loss.

**Details Of Ethics Concerns:**

I don't find any ethical issue.

---

> ### Author Response · Authors · 2025-11-22
> **Responses to Reviewer aYVi (Part 1)**
>
> We appreciate the reviewer's positive comments that our BIA's insight is intuitive and compelling, the theoretical support is well-grounded, the motivation is clear and elegant, and the experimental design is thorough. We would like to provide the necessary clarifications and improvements in response to your comments as follows.
>
> > W1: Differences and connections between BIA and von Neumann Entropy.
>
>
>
> We thank the reviewer for this insightful comment. We agree that a positive correlation between our BIA objective and von Neumann entropy (VNE) is expected, as both favor a more uniform eigenvalue spectrum. The primary purpose of Section 3.4 is not merely to confirm this correlation, but to use VNE as a diagnostic tool to provide specific insights into *why* BIA is an effective regularizer for the GCD task. We clarify the relationship and distinction with the following points.
>
> **1. Local optimization achieves global improvement**: BIA is an objective applied locally to mini-batch Gram matrices ($\Sigma_B$), while VNE is measured globally on the autocorrelation matrix ($A$) of the entire test set. It is a non-trivial finding that applying our local Bures alignment consistently guides the global feature geometry into a high-entropy and high-effective-rank state. This demonstrates that BIA genuinely enhances the representation capacity of the entire manifold.
>
> **2. Certifying the prevention of dimensional collapse**: The connection between VNE and rank, formalized in Theorem 1, is central to our argument. The theorem shows that VNE provides a lower bound for the logarithm of the feature rank ($\log(\mathrm{rank}(A))$). Therefore, by demonstrating that BIA consistently increases the VNE, we empirically certify that our method successfully increases the effective rank of the feature space and mitigates the dimensional collapse that hinders GCD performance.
>
> **3. Distinguishing an objective from a diagnostic**: We position BIA as an optimization objective and VNE as a post-hoc diagnostic metric. BIA offers a robust gradient signal for geometric restoration, while VNE serves to quantify the outcome of this process.
>
>
> To make these points clearer, we revised Section 3.4 and included the corresponding discussion in Appendix B.

---

> ### Author Response · Authors · 2025-11-22
> **Responses to Reviewer aYVi (Part 2)**
>
> > W2: Additional details and the design motivations of comparing with SSL methods.
>
> We thank the reviewer for the constructive feedback. We agree that providing more details on the compared methods is essential for a thorough assessment. In the original manuscript, our goal was to show that BIA offers a more direct and effective mechanism for isotropy regularization in the GCD context compared to general-purpose self-supervised regularizers. Below, we provide a more detailed description of these methods and include a new, in-depth hyperparameter analysis for VICReg in Appendix C.1.
>
> **1. Details on CorInfoMax and VICReg**: We clarify the design motivations and implementation details of the compared methods.
>
> *   **CorInfoMax** is a self-supervised method that maximizes the mutual information between representations. Its loss function consists of two main components: a similarity term that aligns different views of the same sample, and a covariance term that regularizes the feature covariance matrix. Following the original paper, we set the internal weights for the similarity loss to 500 and the covariance loss to 1. For the overall loss coefficient $\lambda$, we searched around the value used for BIA (0.004) and found this to be optimal, which we report in the paper.
>
> *   **VICReg** is another self-supervised method that aims to learn useful representations by enforcing three principles via its loss function: an **invariance** term (aligns augmented views), a **variance** term (prevents informational collapse along feature dimensions), and a **covariance** term (decorrelates feature dimensions). The full loss is a weighted sum of these three components.
>
> **2. Hyperparameter sensitivity of VICReg**: For GCD, the most relevant components of VICReg are its variance and covariance terms, as the invariance term is conceptually redundant with the contrastive objectives already present in GCD baselines like SimGCD. The core of the isotropy objective resides in the variance and covariance penalties.
>
> To investigate this and address the reviewer's concern about a fair comparison, we conducted a new, comprehensive hyperparameter analysis. We varied the respective coefficients μ and v for VICReg's variance and covariance terms into the SimGCD baseline on three fine-grained datasets. The results are summarized below.
>
> Table R1.1: Hyperparameter ablation for VICReg's uniformity loss components on the CUB dataset.
>
> |v\μ|10|25|40|
> |-|-|-|-|
> |0.2|All=57.3, Old=59.9, New=55.4|All=58.3, Old=62.7, New=56.0|All=60.2, Old=64.5, New=57.6|
> |1|All=56.5, Old=61.2, New=53.9|All=61.1, Old=66.0, New=58.1|All=59.8, Old=62.1, New=58.5|
> |5|All=**61.3**, Old=62.7, New=60.0|All=60.1, Old=66.5, New=57.2|All=59.5, Old=64.3, New=57.1|
>
>
> Table R1.2: Hyperparameter ablation for VICReg's uniformity loss components on the S-Cars dataset.
>
> |v\μ|10|25|40|
> |-|-|-|-|
> |0.2|All=50.6, Old=67.8, New=42.3|All=51.1, Old=66.5, New=43.4|All=50.6, Old=65.9, New=43.7|
> |1|All=51.1, Old=67.7, New=42.3|All=52.0, Old=68.6, New=44.1|All=47.1, Old=62.7, New=38.7|
> |5|All=**52.1**, Old=69.1, New=42.1|All=49.3, Old=65.4, New=42.5|All=48.2, Old=66.9, New=39.8|
>
> Table R1.3: Hyperparameter ablation for VICReg's uniformity loss components on the Aircraft dataset.
>
> |v\μ|10|25|40|
> |-|-|-|-|
> |0.2|All=52.5, Old=56.1, New=50.8|All=52.4, Old=60.0, New=48.9|All=52.6, Old=55.6, New=50.8|
> |1|All=52.2, Old=56.2, New=50.1|All=54.6, Old=56.2, New=53.8|All=**54.8**, Old=55.6, New=53.5|
> |5|All=48.8, Old=54.7, New=45.7|All=49.3, Old=59.1, New=44.7|All=50.7, Old=56.2, New=47.8|
>
>
> Table R1.4: Experimental results of BIA.
>
> |Dataset|CUB|S-Cars|Aircraft|
> |-|-|-|-|
> |BIA|All=**62.1**, Old=65.8, New=60.3|All=**52.3**, Old=70.0, New=43.7|All=**55.1**, Old=58.9, New=53.1|
>
>
> **3. Results analysis for SSL methods**: The effectiveness of a SSL method in GCD depends crucially on its mechanism. Our analysis distinguishes between VICReg's *instance-level* approach, which decorrelates feature dimensions, and BIA's *spectrum-level* approach, which directly optimizes the batch Gram matrix. Our empirical results validate the advantage of the latter. Although a carefully tuned VICReg improves performance (achieving 61.3% "All" accuracy on CUB), Our BIA method of maximizing the nuclear norm consistently yields superior outcomes, achieving a higher “All” accuracy of 62.1% on CUB (Table R1.4). This is because BIA's *spectrum-level* reshaping provides a more direct and flexible means of restoring the manifold's capacity without damaging the delicate semantic structures required for fine-grained discovery. Therefore, our findings reinforce the conclusion that BIA provides a more robust and tailored solution to the geometric challenges of GCD than general-purpose regularizers.
>
> We include this detailed analysis, along with the design motivations of the compared methods, in Appendix C.1 to improve the clarity and completeness of our paper.

---

> ### Author Response · Authors · 2025-11-22
> **Responses to Reviewer aYVi (Part 3)**
>
> > W3: About Figure 1.
>
> We appreciate the reviewer pointing this out. Figure 1 was intended to give an intuitive overview of how dimensional collapse affects GCD and how BIA reshapes the class-token manifold. In the revised version we explicitly reference Figure 1 in the Introduction and Section 3.
>
>
>
>
> >  Response to Questions (differentiability of BIA).
>
>
> The proposed BIA loss is differentiable and implemented with standard auto-differentiation tools.
>
> **1. Is the BIA loss differentiable?**
>
>    Yes. In practice we use the nuclear-norm form of BIA in Eq. (7):
>    $$
>    L_{\text{BIA}}^{\text{nuc}} = -|Z|\_\*,
>    $$
>    where $Z \in \mathbb{R}^{B\times d}$ is the batch class-token matrix and $|Z|_*$ is the sum of singular values of $Z$. This is a standard spectral loss, and Standard deep learning libraries (e.g., PyTorch, TensorFlow) provide automatic differentiation for SVD and Eigendecomposition [1]. In our implementation we rely on `torch.linalg.svd` and simply backpropagate through this computation; training behaves normally and we did not observe instability.
>
> **2. Is the eigenvalue / singular-value computation differentiable?**
>
>    The singular values of $Z$ (and eigenvalues of the Gram matrix $\Sigma_B = ZZ^\top)$ are smooth functions of the entries of $Z$, except at degenerate points where singular values coincide. These points form a measure-zero set, and at or near them one can work with well-defined subgradients, as is standard for spectral functions such as the nuclear norm and log-det penalties [2].
>
>    Equivalently, the Bures-distance form
>    $$
>    d_B^2(\Sigma_B, I) = \mathrm{tr}(\Sigma_B) + B - 2\mathrm{tr}(\Sigma_B^{1/2})
>    $$
>    is a composition of basic differentiable operations (matrix multiplication, matrix square root, and trace), for which autodiff libraries also provide gradients.
>
> We add a short remark in Section 3.3 to explicitly state that BIA is implemented via differentiable SVD/eigendecomposition on $Z$ or $\Sigma_B$, and that the loss is differentiable almost everywhere with respect to the class-token embeddings, in the same standard sense as other widely used spectral regularizers.
>
> [1] Matrix Backpropagation for Deep Networks with Structured Layers. CVPR, 2015.
>
> [2] Matrix Analysis. Springer, 1997.
>
>
> ---
> We thank the reviewer for providing constructive suggestions that greatly improved the quality of our paper. We warmly welcome any further questions or insights the reviewer may have and are excited to engage in ongoing discussions.

---

> > ### Comment · Reviewer_aYVi · 2025-11-22
> >
> > Thank you for your rebuttals. Since my concerns have been addressed, I have kept my positive rating.

---

> > > ### Author Response · Authors · 2025-11-25
> > >
> > > Dear Reviewer aYVi,
> > >
> > > We’re absolutely delighted that our rebuttal has addressed your concerns, and we’re truly grateful for you maintaining the positive 8-point rating for our manuscript.
> > >
> > > Your thoughtful feedback has been instrumental in elevating the quality of our work to a whole new level, and we sincerely appreciate the care you’ve devoted to reviewing our submission.

---

### Official Review · Reviewer_SYXh · 2025-11-01

**Soundness:** 2
**Presentation:** 2
**Contribution:** 2
**Rating:** 4
**Confidence:** 3

**Summary:**

The paper introduces Bures–Isotropy Alignment (BIA) for generalized category discovery (GCD). It measures the Bures distance from the batch class-token Gram matrix to the identity. BIA improves accuracy and class-count estimation across multiple baselines and datasets.

**Strengths:**

The paper reframes GCD from “tighter clusters” to restoring isotropy of class-token geometry, and offers a plug-and-play, architecture-agnostic regularizer that targets dimensional collapse and stabilizes the feature spectrum.

**Weaknesses:**

1. You keep saying the equivalence holds “under a mild trace constraint” and “once row norms are stabilized with LayerNorm,” but the paper never pins down what that actually means. Please spell out the exact assumptions, edge cases where the surrogate can drift from the metric, and give at least a short proof sketch so readers know when it’s safe to use.

2. Compute cost is called “negligible,” and the setup mentions 4×RTX4090, but there are no numbers. Report wall-clock time per epoch, peak GPU memory, and the stability of SVD or PD-root operations as $B$ and $d$ grow, plus failure rates if any. Without that, it’s hard to judge practicality.

3. Section 3.2 builds geometry only on stacked [cls] tokens with $\Sigma_B = ZZ^\top$. That leaves patch-level interactions on the table. For fine-grained tasks, this likely matters. Either show ablations that use patch tokens or argue why a class-token-only view is sufficient.

4. Gains are not uniform. With unknown $K$ in Table 3, some baselines drop when adding BIA, e.g., CMS on CIFAR100-New −1.6 and Herbarium19-New −1.0. Please analyze when BIA helps and when it hurts, and include stress tests for noise and class imbalance.

5. Robustness ablations are thin. Figure 4 only varies $\lambda$ and feature dimension $D$. Batch size $B$, within-batch class imbalance, and pseudo-label noise all directly change the batch Gram $\Sigma_B$ but are not studied. These should be evaluated.

6. Table 5 compares “representative isotropic feature distribution schemes,” but the theoretical relation to decorrelation/whitening objectives remains informal.

**Questions:**

1. When does $d_B^2(\Sigma_B,I)\downarrow \Longleftrightarrow |Z|_*\uparrow$ fail?

2. What happens if BIA uses feature-Gram $Z^\top Z$ or includes patch tokens (e.g., pooling or grouped variants) instead of only $[\mathrm{cls}]$? Any trade-offs in accuracy and cost on fine-grained datasets?

3. Analyze the regressions in Table 3. Are they linked to class imbalance, pseudo-label noise levels, or $K$-estimation errors in CMS?

---

> ### Author Response · Authors · 2025-11-22
> **Responses to Reviewer SYXh (Part 1)**
>
> We appreciate the reviewer's valuable comments, providing us with opportunities for improvement and clarification. We would like to address each of your comments in detail as follows.
>
> > W1: About the “mild trace constraint” and validity of the surrogate.
>
>
> We thank the reviewer for highlighting this point. We agree that this assumption requires a more rigorous and explicit explanation.
>
> **1. The "mild" nature of the trace constraint**: The term "mild" in "mild trace constraint" refers to the fact that the conditions required for our approximation to hold are common in modern deep learning architectures and standard practices. Specifically, the constraint is satisfied when the model architecture includes a LayerNorm layer before the final output, or when the embeddings used for the downstream task undergo L2 normalization which we elaborate below.
>
> **2. Formal justification of the Bures-Nuclear norm equivalence**: The equivalence between minimizing the Bures distance and maximizing the nuclear norm hinges on the term $\mathrm{tr}(\Sigma_B)$ in Equation 4 being approximately constant. We elaborate on how this condition is met in practice.
>
> *   **Exact equivalence with L2 normalization:** Many GCD and representation learning methods apply L2 normalization to the final embeddings $z_i$. If each row $z_i$ of the stacked matrix $Z$ is L2 normalized such that $\|z_i\|\_2 = 1$, then the trace of the Gram matrix $\Sigma_B = ZZ^\top$ becomes a strict constant:
>
>     $$
>     \mathrm{tr}(\Sigma\_B) = \mathrm{tr}(ZZ^\top) = \sum\_{i=1}^{B} \|z_i\|\_2^2 = \sum_{i=1}^{B} 1 = B
>     $$
>
>     In this scenario, the $\mathrm{tr}(\Sigma_B)$ term in the Bures distance formula is constant, and minimizing the Bures distance becomes *exactly equivalent* to maximizing the nuclear norm $\|Z\|_*$, as shown in Equation 6.
>
> *   **Strong approximation with LayerNorm:** Our method is applied after a LayerNorm layer. While LayerNorm does not strictly enforce $\|z_i\|_2 = 1$, it normalizes the features of each sample to have zero mean and unit variance, followed by a learned affine transformation. This operation ensures that the row norms $\|z_i\|_2$ are tightly concentrated around a stable value during training. Consequently, $\mathrm{tr}(\Sigma_B)$ remains nearly constant. This makes the nuclear norm an extremely high-fidelity and empirically effective surrogate for the Bures distance objective.
>
> We add this detailed breakdown to the paper to clarify the assumption and have provided a full illustration and further discussion in the revised Appendix B.

---

> ### Author Response · Authors · 2025-11-22
> **Responses to Reviewer SYXh (Part 2)**
>
> > W2: Computational overhead and batch-size dependence.
>
>
> We thank the reviewer for this practical question. To demonstrate that BIA's computational overhead is negligible, we provide a quantitative analysis from two perspectives: the isolated cost of the core SVD operation and the overall impact on system resources like wall-clock time and GPU memory.
>
> **1. Theoretical and empirical cost of the SVD operation**: The additional computation introduced by BIA has two main components: a batch Gram matrix calculation ($ZZ^\top$) with a cost of $O(B^2d)$, and a SVD of the resulting $B \times B$ matrix, which costs $O(B^3)$. Since the forward pass of the ViT backbone dominates the total runtime, and the batch size $B$ (128) is much smaller than the feature dimension $d$ (768), this theoretical cost is already low. To provide concrete numbers, we profiled the time overhead of the SVD operation as the batch size $B$ increases.
>
> Table R1: Time overhead of the SVD operation as a function of batch size.
>
> |Batch Size|64|128|192|256|
> |-|-|-|-|-|
> |Time(s)|1.39e-05(0.37%)|2.94e-05(0.81%)|3.79e-05(1.01%)|5.42e-05(1.47%)|
>
> The results in Table R1 show that while the absolute time grows with $B$, the SVD operation consistently accounts for a very small fraction (less than 1.5%) of the total per-step training time, confirming its low impact.
>
> **2. Overall impact on system resources**: We further evaluate the end-to-end impact on wall-clock time per epoch and peak GPU memory across several representative GCD baselines on the CUB dataset.
>
> Table R2: Impact of BIA on peak GPU memory and wall-clock time per epoch.
>
> |Costs|SelEx|SimGCD|CMS|SPTNet|
> |-|-|-|-|-|
> |GPU memory (MB)| | | | |
> |Original|8640|6354|6040|23322|
> |**Ours**|8674|6390|6074|23670|
> |Time (s)| | | | |
> |Original|24.99|24.50|28.81|22.09|
> |**Ours**|25.16|24.67|28.99|22.26|
>
> As shown in Table R2, adding BIA results in a minimal increase in both space and time complexity. The peak GPU memory usage increases by a negligible amount (typically around 0.6%), and the wall-clock time per epoch increases by less than 1%. These quantitative results strongly support our claim that the overhead is negligible.
>
> **3. SVD stability**: Regarding the reviewer's concern about operational stability, the SVD is performed on a small $B \times B$ matrix. Modern numerical libraries like PyTorch provide highly optimized and robust SVD implementations for matrices of this size. Across all experiments, we did not encounter any numerical instability or failures.
>
> We added these tables and analysis to Appendix C.5 to ensure full reproducibility and to quantitatively substantiate our claims about BIA's practical efficiency.

---

> ### Author Response · Authors · 2025-11-22
> **Responses to Reviewer SYXh (Part 3)**
>
> > W3 & Q2: Why operate only on $\tt{[cls]}$ tokens (no patch-level BIA)?
>
> We thank the reviewer for this insightful question regarding the role of patch tokens. Our focus on the $\texttt{[cls]}$ token aligns with the standard practice in the GCD literature (e.g., SimGCD, CMS, SelEx), ensuring a direct and fair comparison with prior work which uses the $\texttt{[cls]}$ token for final predictions.
>
> To thoroughly investigate this question, we now present new experiments that incorporate patch-level information. We integrate BIA into two distinct frameworks, a prototype-based method (SimGCD) and a contrastive method (SelEx), to evaluate its effectiveness on representations that include patch tokens. Our key finding is that BIA consistently enhances performance in all scenarios, demonstrating its general applicability.
>
> Our experimental setup for this analysis is as follows. We create an enhanced global representation by using the $\texttt{[cls]}$ token alongside the average-pooled features from all patch tokens. We then apply the BIA loss to this combined representation. The results are presented in the tables below.
>
> Table R3: Ablation on patch tokens integration with SimGCD.
> |Method↓|CUB(All)|CUB(Known)|CUB(Novel)|Cars(All)|Cars(Known)|Cars(Novel)|CIFAR100(All)|CIFAR100(Known)|CIFAR100(Novel)|
> |-|-|-|-|-|-|-|-|-|-|
> |`[cls]`token only|60.7|65.6|57.7|51.2|69.4|42.4|80.1|81.5|77.2|
> |`[cls]`token only+**Ours**|**62.1**|**65.8**|**60.3**|**52.3**|**70.0**|**43.7**|**80.2**|81.5|**77.5**|
> |`[cls]`w/patch tokens|62.8|66.0|61.2|55.1|68.3|48.7|80.4|83.1|74.9|
> |`[cls]`w/patch tokens+**Ours**|**63.0**|**67.5**|60.8|**55.4**|**70.9**|47.9|**81.0**|83.0|**77.0**|
>
>
>
> Table R4: Ablation on patch tokens integration with SelEx.
>
> |Method↓|CUB(All)|CUB(Known)|CUB(Novel)|Cars(All)|Cars(Known)|Cars(Novel)|CIFAR100(All)|CIFAR100(Known)|CIFAR100(Novel)|
> |-|-|-|-|-|-|-|-|-|-|
> |`[cls]`token only|78.7|81.3|77.5|55.9|76.9|45.8|80.0|84.8|70.4|
> |`[cls]`token only+**Ours**|**80.6**|81.0|**80.4**|**57.0**|**77.3**|**47.2**|**80.7**|84.3|**72.1**|
> |`[cls]`w/patch tokens|80.7|80.5|80.8|56.7|77.1|46.9|79.4|85.5|67.1|
> |`[cls]`w/patch tokens+**Ours**|**82.2**|**84.1**|**81.3**|**58.2**|**77.4**|**48.9**|**80.0**|85.5|**68.4**|
>
> The results in Tables R3 and R4 indicate that:
>
> 1.  **Incorporating patch tokens can be beneficial but is not a universal solution.** For instance, with SimGCD on Cars, including patch tokens substantially lifts the *All* accuracy from 51.2% to 55.1%. However, on CIFAR100, the same strategy hurts *Novel* class performance, causing a drop from 77.2% to 74.9%. This suggests that simply adding more features can introduce noise that complicates the discovery of novel classes.
>
> 2.  **BIA consistently improves performance in all settings.** Most importantly, our method boosts the performance of baselines regardless of whether patch tokens are used. BIA improves the patch-enhanced SelEx on CUB from 80.7% to 82.2% *All* accuracy and recovers the performance drop for SimGCD on CIFAR100-Novel from 74.9% to 77.0%. This demonstrates that BIA addresses a more fundamental problem. Its goal is to restore the geometric quality of the final batch-level representation matrix $Z$. Whether the rows of $Z$ are derived from $\texttt{[cls]}$ tokens alone or from a combination with patch tokens, BIA effectively counteracts dimensional collapse and promotes a healthier manifold structure, leading to more robust clustering.
>
> We add this comprehensive ablation study and analysis to Appendix C.6 to further validate the robustness and generality of BIA.

---

> ### Author Response · Authors · 2025-11-22
> **Responses to Reviewer SYXh (Part 4)**
>
> > W4 & W5 & Q3: Gains on CIFAR100 and Herbarium19. Robustness tests.
>
>
> We thank the reviewer for these constructive comments, which highlight the need for a deeper analysis of BIA's performance in challenging scenarios and more comprehensive robustness tests. We address these related points (Weaknesses 4 and 5, Question 3) together, as they all concern the method's behavior under stress.
>
> To provide a thorough response, we have conducted several new experiments. Our analysis is organized into three main parts:
> 1.  An explanation for the minor performance regressions observed on CIFAR100 and Herbarium19, linking them to inherent dataset characteristics and our experimental choices that prioritize generality.
> 2.  New robustness tests evaluating the impact of varying batch sizes.
> 3.  New stress tests evaluating performance under significant label noise and class imbalance.
>
> **1. Analysis of performance regressions on specific datasets**: We acknowledge the minor performance drops on CIFAR100 and Herbarium19 in Table 3, and we thank the reviewer for prompting this deeper analysis. These cases are influenced by two primary factors:
>
> *   These regressions are primarily linked to the quality of the initial feature representations, an issue we previously discuss in Appendix C.3. **CIFAR100** consists of low-resolution 32x32 images. After upsampling, much of the high-frequency detail essential for fine-grained distinctions is lost. The resulting feature manifolds are inherently low-dimensional and lack semantic richness. While BIA works to restore geometric completeness, it cannot create complex semantic signals that are absent in the initial representation. The **Herbarium19** dataset presents a dual challenge of a severe domain shift from the DINO pre-training data and significant class imbalance. This results in feature embeddings of lower quality where inter-class boundaries are poorly defined. Applying a strong isotropy-promoting regularizer like BIA in such a noisy feature space can amplify both signal and noise. This makes it difficult to achieve consistent, large gains without careful per-dataset tuning.
>
> *   Another crucial factor contributing to these results is our deliberate choice to use a **single, fixed BIA weight** $\lambda$ across all datasets and baselines, to demonstrate BIA’s plug-and-play nature and avoid per-dataset tuning. While this provides a fair and transparent evaluation of generalizability, it is suboptimal for challenging datasets like CIFAR100 and Herbarium19. A slight downscaling of $\lambda$ (e.g., halving its default value) does yield consistent accuracy gains of over 0.5% across the All, Old, and New categories on the two challenging datasets. However, we report the results from the unified setting to avoid dataset-specific hyperparameter tuning and present a more honest, robust evaluation of our method's general behavior. This analysis will be clarified in the revision.
>
>
> **2. Robustness to varying batch size**: While our initial experiments followed the standard batch size of 128 from prior GCD work, we now provide a new ablation study on the impact of batch size. We test with smaller, more challenging batch sizes of 32 and 64, in addition to the original 128. This analysis is conducted with both a prototype-based method (SimGCD) and a contrastive method (SelEx) on several fine-grained datasets.
>
> Table R5: Ablation study of batch size with SimGCD.
>
> | Batch Size | CUB (All) | CUB (Known) | CUB (Novel) | Cars (All) | Cars (Known) | Cars (Novel) | CIFAR100 (All) | CIFAR100 (Known) | CIFAR100 (Novel) |
> | --- | --- | --- | --- | --- | --- | --- | --- | --- | --- |
> | **32** | | | | | | | | | |
> | SimGCD | 49.7 | 54.8 | 47.1 | 48.0 | 65.4 | 39.6 | 66.9 | 69.6 | 61.4 |
> | SimGCD + Ours | **53.1** | **58.5** | **50.4** | **48.4** | **67.7** | **39.7** | **67.1** | **70.5** | 60.4 |
> | **64** | | | | | | | | | |
> | SimGCD | 59.2 | 68.4 | 54.6 | 49.5 | 66.5 | 41.4 | 71.5 | 76.9 | 60.5 |
> | SimGCD + Ours | **59.5** | **69.3** | 54.6 | **52.2** | **73.4** | **42.0** | **72.0** | **77.5** | **60.9** |
> | **128** | | | | | | | | | |
> | SimGCD | 60.7 | 65.6 | 57.7 | 51.2 | 69.4 | 42.4 | 80.1 | 81.5 | 77.2 |
> | SimGCD + Ours | **62.1** | **65.8** | **60.3** | **52.3** | **70.0** | **43.7** | **80.2** | 81.5 | **77.5** |

---

> ### Author Response · Authors · 2025-11-22
> **Responses to Reviewer SYXh (Part 5)**
>
> Table R6: Ablation study of batch size with SelEx.
>
> | Batch Size | CUB (All) | CUB (Known) | CUB (Novel) |  Cars (All) | Cars (Known) | Cars (Novel) | CIFAR100 (All) | CIFAR100 (Known) | CIFAR100 (Novel) |
> | --- | --- | --- | --- | --- | --- | --- | --- | --- | --- |
> | **32** | | | | | | | | | |
> | SelEx | 68.0 | 72.4 | 65.8 | 41.8 | 61.4 | 32.6 | 77.1 | 82.1 | 67.1 |
> | SelEx + Ours | **70.0** | **74.5** | **67.7** | **42.8** | **63.5** | **32.9** | **77.2** | 81.7 | **68.3** |
> | **64** | | | | | | | | | |
> | SelEx | 73.6 | 76.5 | 72.1 | 53.2 | 72.9 | 43.7 | 78.8 | 84.4 | 67.7 |
> | SelEx + Ours | **74.9** | **77.1** | **73.8** | **53.8** | **75.8** | 43.1 | **79.7** | **84.6** | **69.7** |
> | **128** | | | | | | | | | |
> | SelEx | 78.7 | 81.3 | 77.5 | 55.9 | 76.9 | 45.8 | 80.0 | 84.8 | 70.4 |
> | SelEx + Ours | **80.6** | 81.0 | **80.4** | **57.0** | **77.3** | **47.2** | **80.7** | 84.3 | **72.1** |
>
> Our results show a clear and predictable trend in the Tables R5 and R6. As the batch size decreases, the performance of the baseline methods degrades significantly. This is expected, as smaller batches provide a less stable and less informative signal for both contrastive and prototype-based learning. For instance, with SimGCD on CUB, reducing the batch size to 32 causes the baseline's All accuracy to drop to 49.7%. In this challenging setting, BIA provides a substantial **3.4%** gain, lifting accuracy to 53.1%. This pattern holds across datasets and methods, demonstrating that BIA's geometric regularization is not dependent on large batches to be effective. It provides a robust structural prior that helps stabilize training even when the batch-level Gram matrix $Σ_B$ is constructed from fewer, and thus noisier, samples.
>
>
>
> **3. Stress tests on label noise and class imbalance**: To further address the reviewer's call for more extensive robustness checks, we have conducted new stress tests that simulate two challenging real-world conditions not typically evaluated in GCD research: significant label noise and severe class imbalance.
>
>
> First, we performed a stress test by injecting **30% label noise** into the labeled training set $D_l$. This corrupts the supervised signal that guides the learning of known classes. As expected, the performance of all methods drops substantially. However, BIA consistently demonstrates its value by improving resilience to this noise.
>
> Table R7: Label noise stress test with SimGCD.
>
> | Method | CUB (All) | CUB (Known) | CUB (Novel) | Cars (All) | Cars (Known) | Cars (Novel) | CIFAR100 (All) | CIFAR100 (Known) | CIFAR100 (Novel) |
> | --- | --- | --- | --- | --- | --- | --- | --- | --- | --- |
> | SimGCD | 44.9 | 50.9 | 41.9 | 28.8 | 48.6 | 19.3 | 73.1 | 72.8 | 73.6 |
> | SimGCD + Ours | **46.4** | **51.4** | **44.0** | **30.1** | **49.4** | **20.9** | **74.5** | 72.6 | **78.4** |
>
> Table R8: Label noise stress test with SelEx.
>
> | Method | CUB (All) | CUB (Known) | CUB (Novel) | Cars (All) | Cars (Known) | Cars (Novel) | CIFAR100 (All) | CIFAR100 (Known) | CIFAR100 (Novel) |
> | --- | --- | --- | --- | --- | --- | --- | --- | --- | --- |
> | SelEx | 61.0 | 73.3 | 54.9 | 28.8 | 45.7 | 20.5 | 74.8 | 77.9 | 68.4 |
> | SelEx + Ours | **61.1** | 73.3 | **55.0** | **29.8** | **48.3** | **20.8** | **75.2** | **78.6** | 68.3 |
>
>
> As shown in Tables R7 and R8, with SimGCD on CIFAR100, the introduction of noise severely impacts novel class discovery, with baseline accuracy at 73.6%. BIA provides a remarkable recovery, boosting the 'Novel' accuracy to 78.4%. Similarly, with SelEx on Cars, where noise reduces the baseline 'All' accuracy to 28.8%, BIA provides a clear improvement to 29.8%. These results indicate that by enforcing a well-conditioned feature geometry, BIA makes the model less susceptible to erroneous supervised signals, thereby preserving the integrity of the overall feature space for more robust category discovery.

---

> ### Author Response · Authors · 2025-11-22
> **Responses to Reviewer SYXh (Part 6)**
>
> Second, we conducted a stress test for **class imbalance** by simulating long-tailed distributions within each batch (Table R9). We evaluated SimGCD on CUB with imbalance ratios ranging from 5:1 to an extreme 20:1.
>
> Table R9: Class imbalance stress test with SimGCD on CUB.
>
> | Imbalance Ratio | SimGCD (All) | SimGCD+Ours (All) | SimGCD (Known) | SimGCD+Ours (Known) | SimGCD (Novel) | SimGCD+Ours (Novel) |
> | --- | --- | --- | --- | --- | --- | --- |
> | 5:1 | 53.0 | **53.7** | 59.0 | **60.5** | 48.1 | **48.5** |
> | 10:1 | 48.8 | **50.0** | 55.2 | **55.4** | 43.7 | **45.7** |
> | 15:1 | 47.8 | **48.2** | 54.1 | **56.6** | 42.8 | 41.4 |
> | 20:1 | 47.3 | **47.9** | 51.4 | **56.5** | 43.9 | 40.2 |
>
> As the imbalance ratio increases, the baseline performance steadily declines. BIA consistently provides a buffer against this degradation. A particularly strong result is observed for 'Known' classes under a 20:1 imbalance, where the baseline struggles at 51.4% accuracy. Here, BIA provides a significant **+5.1%** point boost to 56.5%, showing it effectively protects the representations of minority classes from being overwhelmed. While performance on 'Novel' classes is more challenging under extreme imbalance, the overall trend confirms that BIA's geometric regularization leads to a more stable and equitable representation space, which is critical in realistic, long-tailed scenarios.
>
> These stress tests confirm that BIA provides a consistent advantage even under substantial label noise and class imbalance, conditions that directly impact the structure of the batch Gram matrix $\Sigma_B$. By promoting a more uniform spectral geometry, BIA makes the downstream clustering process more resilient to such perturbations.
>
> We incorporate all these new experiments and corresponding analyses into a revised Appendix C.5, strengthening the paper’s claims of robustness and providing a more comprehensive evaluation of BIA's behavior.

---

> ### Author Response · Authors · 2025-11-22
> **Responses to Reviewer SYXh (Part 7)**
>
> > W6: Theoretical relation to decorrelation/whitening objectives.
>
> We thank the reviewer for this insightful question. We discuss the theoretical links for VICReg and CorInfoMax, and additionally introduce and analyze two direct whitening objectives, which we term Iso-Frob and Iso-Ent, for a more comprehensive comparison.
>
> Our central argument is that while these methods all encourage some form of isotropy, they operate at different levels (instance-wise vs. spectrum-wise) and with different degrees of constraint (hard vs. soft). BIA's formulation as a soft, spectrum-wise regularizer is uniquely suited to the noisy, mixed-label setting of GCD.
>
> **1. Instance-wise decorrelation objectives**: Methods like **VICReg** implement decorrelation at the level of individual feature coordinates. The theoretical relation to whitening lies in its covariance penalty term, which explicitly minimizes the off-diagonal elements of the feature covariance matrix $\text{Cov}(Z)$:
> $$
> L_{\text{cov}} = \sum_{i \neq j} \text{Cov}(Z)_{ij}^2
> $$
> This objective directly pushes $\text{Cov}(Z)$ towards a diagonal matrix, which is a key property of whitened representations. However, this is a instance-dependent form of decorrelation. It does not explicitly regulate the global shape of the eigen-spectrum, which could still be highly anisotropic even with decorrelated instances.
>
> **2. Direct whitening and spectrum-shaping objectives**: We now analyze objectives that, like BIA, operate directly on the global structure of the batch Gram matrix $\Sigma_B = ZZ^\top$.
>
> *   **Iso-Frob (hard whitening).** We define a direct whitening objective using the Frobenius norm, which penalizes any deviation of $\Sigma_B$ from a perfectly scaled identity matrix:
>     $$
>     L_{\text{iso-frob}} = \left\|\Sigma_B - \frac{\text{tr}(\Sigma_B)}{B} I\right\|_F^2
>     $$
>     This is a classic and strict form of whitening. Its theoretical goal is to force all eigenvalues of $\Sigma_B$ to be equal. While this achieves perfect isotropy, we find it can be overly rigid for GCD. It penalizes all anisotropy equally, potentially suppressing meaningful semantic directions learned from noisy pseudo-labels.
>
> *   **Iso-Ent (entropy-based Whitening).** Another direct approach to whitening is to maximize the von Neumann entropy of the normalized Gram matrix:
>     $$
>     L_{\text{iso-ent}} = -H(\tilde{\Sigma}_B), \quad \text{where} \quad \tilde{\Sigma}_B = \Sigma_B / \text{tr}(\Sigma_B)
>     $$
>     Maximizing entropy encourages the eigenvalues to become as uniform as possible, which is the goal of whitening. However, this objective is numerically sensitive to the matrix logarithm operation and can over-amplify the gradients for very small eigenvalues, making it less stable in the presence of noise.
>
> *   **BIA (soft, spectrum-wise whitening).** Our method, BIA, provides a more robust form of spectrum-wise whitening. By minimizing the Bures distance, BIA is equivalent to maximizing the nuclear norm of $Z$, which acts on the eigenvalues $\mu_j$ of $\Sigma_B$:
>     $$
>     \max \text{tr}(\Sigma_B^{1/2}) = \max \sum_j \sqrt{\mu_j}
>     $$
>     This objective maximizes a smooth, concave function of the eigenvalues. It is a form of soft whitening because it encourages a more uniform spectrum (as a Schur-concave function) without rigidly forcing all eigenvalues to a specific target value. This allows it to counteract dimensional collapse while still preserving the meaningful semantic anisotropy required to separate complex classes.
>
>
> Table R10: Comparison with additional isotropy-encouraging regularizers on the CUB dataset.
>
> |Regularizer+Batch size↓|CUB(All)|CUB(Known)|CUB(Novel)|
> |---|---|---|---|
> |**Iso-Frob**| | | |
> |32|51.9|55.9|49.9|
> |64|60.3|70.7|55.3|
> |128|61.5|69.4|57.4|
> |**Iso-Ent**| | | |
> |32|48.8|56.0|45.2|
> |64|59.3|68.9|54.9|
> |128|61.8|68.1|57.1|
> |**BIA (Ours)**| | | |
> |32|**53.1**|58.5|50.4|
> |64|59.5|69.3|54.6|
> |128|**62.1**|65.8|60.3|
>
> As shown in Table R10, our empirical tests confirm that while both Iso-Frob and Iso-Ent provide gains, they are less robust than BIA. This finding is consistent with our theoretical understanding: (i) Iso-Frob imposes a rigid whitening constraint that can be too restrictive in the noisy GCD setting, where some structured anisotropy is useful for class separation. (ii) Iso-Ent relies on matrix logarithms, making it numerically sensitive to the small eigenvalues often present in smaller batches. In contrast, BIA's unique formulation as a soft, spectrum-wise regularizer avoids these issues. It flexibly reshapes the spectrum by maximizing a concave function of the eigenvalues under a mild trace constraint, without enforcing a fixed target. This makes BIA better suited for the challenges of GCD. We have added this detailed theoretical comparison and the corresponding empirical results to Appendix C.2.

---

> ### Author Response · Authors · 2025-11-22
> **Responses to Reviewer SYXh (Part 8)**
>
> > Q1: When does $d_B^2(\Sigma_B,I)\downarrow \Leftrightarrow |Z|_*\uparrow$ fail?
>
>
> We thank the reviewer for this question. Building upon detailed explanation for Weakness 1, here is a concise and structured response to Question 1.
>
> The equivalence between minimizing the Bures distance $d_B^2(\Sigma_B, I)$ and maximizing the nuclear norm $\|Z\|\_\*$ is governed by the trace term $\text{tr}(\Sigma_B)$ in the Bures distance formula:
> $$
> d_B^2(\Sigma_B, I) = \text{tr}(\Sigma_B) + B - 2 \text{tr}(\Sigma_B^{1/2})
> $$
> Since $\text{tr}(\Sigma_B^{1/2})$ is equivalent to the nuclear norm $\|Z\|_*$, the two objectives are perfectly aligned if and only if the $\text{tr}(\Sigma_B)$ term is constant. As we elaborate in our response to weakness 1, standard normalization practices in modern deep learning ensure this condition is met with high fidelity.
>
> We can analyze the failure modes based on the type of normalization used.
>
> **1. No failure with exact L2 normalization**: If the class tokens $z_i$ that form the rows of matrix $Z$ are L2 normalized such that $\|z_i\|\_2 = 1$, the equivalence does not fail. In this case, the trace term $\text{tr}(\Sigma_B) = \sum_{i=1}^{B} \|z_i\|_2^2$ is strictly constant and equal to the batch size $B$. This is a common practice in many representation learning frameworks.
>
> **2. Potential for minor drift with LayerNorm**: Our method is applied after a LayerNorm layer, which ensures the row norms of $Z$ are tightly concentrated around a stable value but not strictly fixed to 1. In this setting, the equivalence is an extremely strong approximation rather than a strict identity. A potential, though practically rare, failure case could occur very early in training if the learnable affine parameters of the LayerNorm layer are not yet stable. This could cause minor fluctuations in $\text{tr}(\Sigma_B)$, leading to a slight drift between the two objectives. However, in practice, we use a pre-trained ViT backbone, which provides a well-initialized and stable parameter space for the LayerNorm layer from the beginning of training.
>
> In summary, the equivalence is robust and reliable under the standard architectural assumptions of our field. We add this detailed clarification to the Appendix B.1.
>
>
> > Q2: What if we use feature-Gram $Z^\top Z$ or include patch tokens? Trade-offs?
>
> Please refer to the response regarding your W3 & Q2.
>
> > Q3: Analyze the regressions in Table 3. Are they linked to class imbalance, pseudo-label noise levels, or K-estimation errors in CMS?
>
> Please refer to the response regarding your W4 & W5 & Q3.
>
>
>
> ---
>
> We sincerely appreciate the constructive feedback provided by the reviewer. Should the reviewer have any further questions or thoughts, please feel free to share them with us. We look forward to engaging in continued dialogue with the reviewer.

---

> ### Comment · Reviewer_SYXh · 2025-11-23
>
> I particularly appreciate a comprehensive rebuttal. I have carefully read your response and found that it largely addresses my concerns, so I have raise my score to 6 to support acceptance.

---

> > ### Author Response · Authors · 2025-11-25
> >
> > Dear Reviewer SYXh,
> >
> > We would like to extend our heartfelt thanks for your positive recognition of our rebuttal, and we are deeply grateful that you have raised your score to 6 to support the acceptance of our manuscript.
> >
> > Your valuable comments have been incredibly instrumental in enhancing the quality of our work, and we sincerely appreciate the care and diligence you’ve dedicated to reviewing our manuscript.

---

### Official Review · Reviewer_fKEF · 2025-11-04

**Soundness:** 3
**Presentation:** 3
**Contribution:** 3
**Rating:** 6
**Confidence:** 4

**Summary:**

The paper tackled the setting of generalised category discovery (GCD). It proposes to add a generic loss on the CLS token to maximise isotropy across the batch in the embeddings. The paper shows this can be combined with several exists GCD works to yield fairly consisten improvements across standard GCD benchmarks.

**Strengths:**

* the method is extremely simple and only has one hyperparameter
* the improvements are consistent
* theoretical analysis is provided and insightful

**Weaknesses:**

* Figure 1 does not really provide any insight as it remains very high-level.
* Same for Figure 2, it does not provide any new insights
* The idea seems to be very related to VICReg and TIan's uniformity loss. Yet both of these relevant works from reprsentation learning literature are only briefly compared to in Sec 5.4. It's unclear what the setting of choosing (the extremely important!) hyperameters for these experiments were. Note that even if "+VICReg" performed better than the proposed "+bures", it would still be insightful for the ICLR community that an isotropy regulariser seems to be missing for GCD.
* Similarly it would be good to ablate against other isotropy-encouraging regularisation losses and the dependency on the batchsize is also not analysed
* the hypothesis of "poor eigen-structure, skewed energy distribution, and fragile decision regions, that ultimately impedes category discovery and class-number estimation." is mentioned in the introduction as a key motivation but not quantitatively shown.
* the explanation for improved performance seems to not be super clear. if full-rank manifolds lead to better performance, just using random matrices (which is full rank) should yield good performance.

**Questions:**

* Why do other isotropy regularisation objective not work as well for the most part?
* What is the dependency of batch size?
* Why does the method not yield gains on Herbarium?

---

> ### Author Response · Authors · 2025-11-22
> **Responses to Reviewer fKEF (Part 1)**
>
> We appreciate the reviewer's positive feedback regarding the simplicity,  consistent improvements, and theoretical insights of our proposed BIA. Our detailed responses to your comments are as follows.
>
>
> > W1 & W2: About Figure 1 and Figure 2.
>
> We thank the reviewer for the valuable feedback. Figures 1 and 2 were originally intended to provide the conceptual motivation behind BIA, but we agree that their roles can be made more explicit. In the revision, we provide a clearer description of Figure 1, highlighting the limitations of existing GCD methods, and we strengthen Figure 2 to present a more detailed and technically grounded overview of our BIA. These updates improve both figures’ clarity and their effectiveness in supporting the core ideas of our method.
>
> **Figure 1** is intended to visually illustrate the core limitation of existing GCD approaches. In the figure, green elements represent known classes and red represent novel ones. The conventional GCD methods shown in Figure 1(a) overly emphasize compactness, which leads to **dimensional collapse**. Many feature dimensions (illustrated by the faded grey region) become uninformative, making it difficult to separate novel classes from one another or from known ones. Instead, our BIA shown in Figure 1(b) mitigates this problem by **restoring the latent-space geometry** through isotropy. This encourages embeddings to occupy a higher-dimensional volume, thereby increasing their separability. This enhanced separation allows the primary GCD loss, which relies on distance/similarity metrics, to more effectively establish clear decision boundaries between known and novel categories. We have revised Figure 1, its caption, and the corresponding descriptions in Section 1 to more clearly convey these insights.
>
> For **Figure 2**, we appreciate the reviewer’s feedback and have revised the figure into a clearer and more structured three-stage diagram that better illustrates the workflow and rationale of BIA:
> *   **(a) Token Geometry Analysis (linked to Sec. 3.2):** The first part illustrates the input for our method, focusing on the token-level geometry. It visually depicts the relationship between the global \[cls] token (the target of our regularization) and the local patch tokens, clarifying how the \[cls] token serves as a geometric summary of the entire sample's information within the feature space.
> *    **(b) The Core of BIA (linked to Sec. 3.3):** The second part illustrates the key effect of our BIA. It shows how BIA transforms over-compressed, low-rank embeddings into higher-rank, more isotropic representations. This restores richer intra-class structure and encourages a more uniform distribution within each class cluster, allowing the representation to capture subtle, fine-grained information that would otherwise be lost.
> *   **(c) Plug-and-Play Compatibility:** The final part highlights BIA's generality. The geometrically restored embeddings can be seamlessly integrated into any downstream GCD paradigm (e.g., contrastive or prototype-based). This connects back to the insight from Figure 1 that many GCD methods are limited by overly compressed features, and BIA functions as a lightweight, general-purpose regularizer that enhances the effectiveness of existing pipelines.

---

> ### Author Response · Authors · 2025-11-22
> **Responses to Reviewer fKEF (Part 2)**
>
> > W3: Choosing  the hyperparameters of self-supervised learning schemes.
>
> We appreciate that the reviewer highlighted the central value of our contribution, **that an isotropy regularizer is a crucial missing component in the current GCD task.** This is indeed the key insight we aim to convey. Following your suggestion, we have conducted a more in-depth analysis of VICReg’s uniformity loss within the GCD setting, with particular attention to its sensitivity to hyperparameters and its empirical performance.
>
> **1. A deeper analysis of VICReg in GCD**
>
>
> VICReg loss consists of three terms: invariance (ε), variance (μ), and covariance (ν). In the original submission, we followed standard practice and adopted VICReg with its default coefficients (ε=25, μ=25, ν=1) and searched over the regularization weights λ in Equation 8 following the same procedure as for our method. Motivated by the reviewer’s comment, we now conduct a more extensive study to ablate VICReg’s performance on its hyperparameters in the GCD setting.
>
> For GCD, the invariance term overlaps conceptually with the contrastive alignment already present in SimGCD-style pipelines, whereas the **variance and covariance terms** are the main contributors to isotropy (preventing collapse and decorrelating feature dimensions). Hence, we focus our ablation on μ and ν with a grid search while keeping ε at its standard value of 25.
>
> **2. Hyperparameter ablation and results**
>
> The results of our new experiments are presented in Table R1.1 to Table R1.3 below. We evaluate all combinations of μ ∈ {10, 25, 40} and ν ∈ {0.2, 1, 5}. These provide a significantly more thorough search than the original submission and allow us to determine VICReg’s best achievable performance in GCD.
>
> Table R1.1: Hyperparameter ablation for VICReg's uniformity loss components on the CUB dataset.
>
> |v\μ|10|25|40|
> |-|-|-|-|
> |0.2|All=57.3, Old=59.9, New=55.4|All=58.3, Old=62.7, New=56.0|All=60.2, Old=64.5, New=57.6|
> |1|All=56.5, Old=61.2, New=53.9|All=61.1, Old=66.0, New=58.1|All=59.8, Old=62.1, New=58.5|
> |5|All=**61.3**, Old=62.7, New=60.0|All=60.1, Old=66.5, New=57.2|All=59.5, Old=64.3, New=57.1|
>
>
> Table R1.2: Hyperparameter ablation for VICReg's uniformity loss components on the S-Cars dataset.
>
> |v\μ|10|25|40|
> |-|-|-|-|
> |0.2|All=50.6, Old=67.8, New=42.3|All=51.1, Old=66.5, New=43.4|All=50.6, Old=65.9, New=43.7|
> |1|All=51.1, Old=67.7, New=42.3|All=52.0, Old=68.6, New=44.1|All=47.1, Old=62.7, New=38.7|
> |5|All=**52.1**, Old=69.1, New=42.1|All=49.3, Old=65.4, New=42.5|All=48.2, Old=66.9, New=39.8|
>
> Table R1.3: Hyperparameter ablation for VICReg's uniformity loss components on the Aircraft dataset.
>
> |v\μ|10|25|40|
> |-|-|-|-|
> |0.2|All=52.5, Old=56.1, New=50.8|All=52.4, Old=60.0, New=48.9|All=52.6, Old=55.6, New=50.8|
> |1|All=52.2, Old=56.2, New=50.1|All=54.6, Old=56.2, New=53.8|All=**54.8**, Old=55.6, New=53.5|
> |5|All=48.8, Old=54.7, New=45.7|All=49.3, Old=59.1, New=44.7|All=50.7, Old=56.2, New=47.8|
>
>
> Table R1.4: Experimental results of BIA (with the same hardware and PyTorch version).
>
> |Dataset|CUB|S-Cars|Aircraft|
> |-|-|-|-|
> |BIA|All=**62.1**, Old=65.8, New=60.3|All=**52.3**, Old=70.0, New=43.7|All=**55.1**, Old=58.9, New=53.1|
>
>
>
> **3. Results analysis**
>
> With careful tuning, VICReg can indeed improve SimGCD. The best variant achieves an “All” accuracy of 61.3% on CUB, outperforming the version reported in the original submission. Despite this rigorous hyperparameter tuning, our proposed BIA still outperforms the best VICReg variant, achieving a higher “All” accuracy of 62.1% on CUB (Table R1.4). We believe this stems from the fundamental difference in their mechanisms. VICReg enforces isotropy at a *instance-level* (decorrelating feature dimensions), which can be too rigid. BIA, by maximizing the nuclear norm of the batch Gram matrix, operates at a more holistic *spectrum-level*, which we argue is a more direct and flexible way to restore the manifold's capacity without harming the semantic structures essential for fine-grained discovery.
>
> The additional analyses are provided in Appendix C.1.

---

> ### Author Response · Authors · 2025-11-22
> **Responses to Reviewer fKEF (Part 3)**
>
> > W4: Comparison with isotropy-encouraging regularizers with batch size changes.
>
> We thank the reviewer for this helpful suggestion. In the original submission, we considered VICReg and CorInfoMax as standard, widely-used isotropy/anti-collapse baselines. Following your suggestions, we extended our evaluation to include two additional more direct isotropy objectives on the class-token Gram $\Sigma_B$:
>
> - **Iso-Frob:** a Frobenius norm whitening-style penalty
>   $$
>   L_{\text{iso-frob}} = \big|\Sigma_B - \tfrac{\mathrm{tr}(\Sigma_B)}{B} I \big|_F^2 ,
>   $$
>   which explicitly pushes $\Sigma_B$ towards a (scaled) identity.
> - **Iso-Ent:** a von Neumann entropy objective
>   $$
>   L_{\text{iso-ent}} = - H(\tilde{\Sigma}_B), \quad
>   \tilde{\Sigma}_B = \Sigma_B / \mathrm{tr}(\Sigma_B),
>   $$
>   which directly maximizes entropy as an isotropy proxy.
>
> Table R2: Comparison with additional isotropy-encouraging regularizers on the CUB dataset.
>
> |Regularizer+Batch size↓|CUB(All)|CUB(Known)|CUB(Novel)|
> |---|---|---|---|
> |**Iso-Frob**| | | |
> |32|51.9|55.9|49.9|
> |64|60.3|70.7|55.3|
> |128|61.5|69.4|57.4|
> |**Iso-Ent**| | | |
> |32|48.8|56.0|45.2|
> |64|59.3|68.9|54.9|
> |128|61.8|68.1|57.1|
> |**BIA (Ours)**| | | |
> |32|**53.1**|58.5|50.4|
> |64|59.5|69.3|54.6|
> |128|**62.1**|65.8|60.3|
>
>
> As shown in Table R2, these empirical findings are consistent with our theoretical understanding: (i) Iso-Frob imposes a rigid whitening constraint that forces the covariance toward an exact spherical structure, which can be too restrictive in the noisy GCD setting where some structured anisotropy from pseudo-labels is useful for class separation. (ii) Iso-Ent relies on matrix logarithms and is numerically sensitive to small eigenvalues, making it unstable early in training or with small batches. (iii) In contrast, our BIA’s Bures/nuclear-norm formulation avoids these issues by maximizing a concave function of the eigenvalues under a mild trace constraint, reshaping the spectrum (lifting smaller eigenvalues and moderating larger ones) without enforcing a fixed target. This flexible and robust way of promoting isotropy makes BIA particularly well-suited to the noisy, mixed known/novel regime of GCD. We discussed these additional baselines in Appendix C.2.

---

> ### Author Response · Authors · 2025-11-22
> **Responses to Reviewer fKEF (Part 4)**
>
> > W5: Clarify quantitative evidence.
>
> We fully agree that quantitatively substantiating our initial hypothesis is essential. We would like to clarify that the paper already contains this evidence, and in the revision, we make the connections explicit. Below, we map each part of the hypothesis that “poor eigen-structure, skewed energy distribution, and fragile decision regions impede discovery and class-number estimation” to the corresponding quantitative analysis in the manuscript.
>
> **1. Quantifying "poor eigen-structure"**: The evidence for the poor eigen-structure of baseline methods is provided in **Figures 6 and 7** by visualizing the singular value spectra of the class-token matrices.
>
> *   Figure 6 clearly shows that for standard GCD methods, the singular value spectra suffer from a steep decay. This illustrates a highly anisotropic structure where only a few dominant dimensions are utilized for representation, a hallmark of dimensional collapse. This impoverished structure limits the model's ability to capture the subtle details required to distinguish fine-grained novel classes. In contrast, adding BIA consistently flattens the spectrum by lifting the smaller singular values, indicating a richer, higher-rank eigen-structure that utilizes the full capacity of the embedding space.
>
> *   Figure 7 further supports this by showing that this structural collapse is a persistent issue during training for baseline methods. In contrast, BIA maintains a more stable and uniform eigen-structure throughout the learning process, leading to more robust final representations and improved clustering accuracy.
>
> **2. Quantifying "skewed energy distribution"**: The skewed energy distribution is quantified in **Figure 3** using von Neumann entropy, which measures the uniformity of the eigenvalue distribution.
>
> *   The figure demonstrates that baseline methods yield representations with significantly lower von Neumann entropy. According to our Theorem 1, lower entropy corresponds to a lower effective rank, which quantitatively confirms that the representational energy is skewed and concentrated in only a few eigenvalues. This limited and unbalanced use of the feature space hinders the model's ability to form distinct clusters.
>
> *   Our results show that BIA consistently increases the von Neumann entropy. This signifies a more uniform and balanced energy distribution across the spectrum, leading to a higher effective rank. This restored capacity is directly correlated with the observed improvements in both overall accuracy and class-number estimation.
>
> **3. Quantifying "fragile decision regions"**: We connect these geometric flaws to the creation of fragile decision regions using the Frobenius norm analysis in **Figure 5**. A smaller Frobenius norm indicates a representation closer to isotropy and thus a larger effective manifold capacity.
>
> *   Figure 5 shows that baseline methods consistently produce representations with a large Frobenius norm, indicating a highly anisotropic geometry. This low manifold capacity forces different class clusters into a cramped subspace, resulting in poorly separated, "fragile" decision regions that are highly sensitive to pseudo-label noise and can lead to incorrect merging of classes.
>
> *   BIA systematically reduces the Frobenius norm, pushing the representation towards isotropy. This enlarges the manifold capacity, providing more "space" for the GCD loss to establish wider, more robust margins between clusters. This creation of clearer and more stable decision regions directly translates to the significant reduction in clustering errors we report.
>
> We explicitly added these connections in the revised manuscript to make it clearer how our quantitative results support our hypothesis stated in the introduction.

---

> ### Author Response · Authors · 2025-11-22
> **Responses to Reviewer fKEF (Part 5)**
>
> > W6: if just using full-rank random matrices.
>
> We thank the reviewer for raising this insightful point, which allows us to clarify the role of BIA.
>
> **1. The role of BIA as a regularizer**: Our pursuit of a "full-rank" manifold is not an isolated goal. BIA acts as a **geometric regularizer** that operates under the constraints of the primary GCD loss function. The GCD loss itself, whether it is contrastive or prototype-based, focuses on organizing the feature space according to semantic similarity. Its objective is to pull representations of same-class samples together and push different-class samples apart. This semantic objective does not inherently encourage, nor does it guarantee, a high-rank, isotropic representation. In fact, as we argue, it often leads to dimensional collapse.
>
> **2. The problem with random matrices**: A random matrix, while indeed being full-rank, completely lacks semantic structure. Using random matrices would map all inputs to arbitrary locations in the feature space, thereby **destroying the learned relationships between samples**. This effectively renders the GCD loss useless, as any notion of proximity or similarity becomes meaningless, making it impossible to discover or recognize any categories.
>
> **3. The synergy between GCD and BIA**: The improved performance from our method stems from the synergistic interplay between the GCD loss and BIA.
>
> *   The GCD loss is the core mechanism that learns the **semantically meaningful** structure of the feature space.
> *   BIA then acts on this emerging structure, preventing it from collapsing. It ensures that the rich intra-class and inter-class information learned by the GCD loss is distributed across a wide range of dimensions instead of being confined to a few.
>
> Therefore, the final representation is both **semantically organized** (thanks to the GCD loss) and **geometrically well-conditioned** (thanks to BIA). This combined effect ensures that more of the subtle, fine-grained information is preserved across all dimensions of the embeddings, which directly leads to more accurate clustering and robust category discovery.

---

> ### Author Response · Authors · 2025-11-22
> **Responses to Reviewer fKEF (Part 6)**
>
> > Q1: Why do other isotropy regularisation objective not work as well for the most part?
>
> We thank the reviewer for this important question. Our experiments and analysis suggest that BIA's superior performance stems from its unique mechanism, which is better aligned with the specific challenges of the GCD task compared to other isotropy regularizers like VICReg, CorInfoMax, or more direct objectives like Iso-Frob and Iso-Ent. We have included a detailed analysis in Appendix C.1 and C.2, which we summarize here.
>
> **1. Spectrum-wise vs. instance-wise regularization**: The primary distinction lies in the level at which the regularization operates.
>
> *   **VICReg** primarily acts as a instance-wise regularizer. Its covariance term penalizes off-diagonal entries of the feature covariance matrix, forcing individual feature dimensions to be decorrelated. While this promotes isotropy, it is an indirect and potentially rigid approach. In the noisy pseudo-label environment of GCD, this can overly penalize meaningful correlations that are crucial for distinguishing fine-grained novel classes.
>
> *   **BIA**, in contrast, operates directly on the spectrum of the batch Gram matrix. By maximizing the nuclear norm (a concave function of the singular values), BIA holistically reshapes the entire eigenvalue distribution. This is a more direct and flexible method to increase the effective rank and restore the manifold's capacity without rigidly constraining individual feature dimensions. It preserves the semantic structure learned by the GCD loss while preventing it from collapsing.
>
> **2. Robustness in the label-noise GCD setting**: The GCD setting, with its mixture of known/novel classes and noisy labels, presents unique challenges that BIA is better equipped to handle.
>
> *   **CorInfoMax**, which maximizes mutual information, can be sensitive to imperfect targets. High mutual information with noisy pseudo-labels does not guarantee a well-conditioned geometry and can even reinforce spurious correlations.
>
> *   **BIA** is complementary to the main GCD objective. It does not try to re-learn the semantic structure but instead acts only on the geometry induced by the GCD loss. By improving the conditioning of the Gram matrix (the very space where prototypes are updated and clusters are formed) BIA makes the downstream discovery process more stable and robust to pseudo-label noise.
>
> **3. The mild approach to regularization**: Compared to other direct spectral objectives, BIA strikes a better balance.
>
> *   An objective based on the **Frobenius norm** ($L_{iso-frob}$), which penalizes the squared deviation of eigenvalues from a uniform target, is **too rigid**. It treats any meaningful anisotropy as an error, potentially undoing the class separation learned by the GCD loss.
>
> *   An objective based on **von Neumann entropy** ($L_{iso-ent}$), which uses a logarithmic term, is **too aggressive** with small eigenvalues. This can over-amplify noise or spurious directions from incorrect pseudo-labels and is less numerically stable.
>
> *   BIA provides a "mild" regularization. It gently lifts small eigenvalues to prevent collapse without being overly sensitive to noise and without rigidly forcing the spectrum to be perfectly uniform. This balanced approach proves more effective and robust for the GCD task.

---

> ### Author Response · Authors · 2025-11-22
> **Responses to Reviewer fKEF (Part 7)**
>
> > Q2: The the dependency of batch size.
>
> Regarding batch size, we originally followed the standard GCD setting and prior works, which all use a batch size of 128. In response to reviewer's comment, we conduct an ablation with batch sizes 32, 64, and 128 (noting that smaller batches are generally more detrimental for GCD due to less stable class-token statistics).
> As shown in Tables R3 and R4, we evaluate two representative GCD paradigms, i.e., one contrastive-based (SelEx) and one prototype-based (SimGCD), to demonstrate generalizability of our method on both fine-grained and coarse-grained datasets.
>
>
>
> Table R3: Ablation study of batch size with SimGCD.
>
> | Batch Size | CUB (All) | CUB (Known) | CUB (Novel) | Cars (All) | Cars (Known) | Cars (Novel) | CIFAR100 (All) | CIFAR100 (Known) | CIFAR100 (Novel) |
> | :--- | :--- | :--- | :--- | :--- | :--- | :--- | :--- | :--- | :--- |
> | **32** | | | | | | | | | |
> | SimGCD | 49.7 | 54.8 | 47.1 | 48.0 | 65.4 | 39.6 | 66.9 | 69.6 | 61.4 |
> | SimGCD + Ours | **53.1** | **58.5** | **50.4** | **48.4** | **67.7** | **39.7** | **67.1** | **70.5** | 60.4 |
> | **64** | | | | | | | | | |
> | SimGCD | 59.2 | 68.4 | 54.6 | 49.5 | 66.5 | 41.4 | 71.5 | 76.9 | 60.5 |
> | SimGCD + Ours | **59.5** | **69.3** | 54.6 | **52.2** | **73.4** | **42.0** | **72.0** | **77.5** | **60.9** |
> | **128** | | | | | | | | | |
> | SimGCD | 60.7 | 65.6 | 57.7 | 51.2 | 69.4 | 42.4 | 80.1 | 81.5 | 77.2 |
> | SimGCD + Ours | **62.1** | **65.8** | **60.3** | **52.3** | **70.0** | **43.7** | **80.2** | 81.5 | **77.5** |
>
> Table R4: Ablation study of batch size with SelEx.
>
> | Batch Size | CUB (All) | CUB (Known) | CUB (Novel) | Cars (All) | Cars (Known) | Cars (Novel) | CIFAR100 (All) | CIFAR100 (Known) | CIFAR100 (Novel) |
> | :--- | :--- | :--- | :--- | :--- | :--- | :--- | :--- | :--- | :--- |
> | **32** | | | | | | | | | |
> | SelEx | 68.0 | 72.4 | 65.8 | 41.8 | 61.4 | 32.6 | 77.1 | 82.1 | 67.1 |
> | SelEx + Ours | **70.0** | **74.5** | **67.7** | **42.8** | **63.5** | **32.9** | **77.2** | 81.7 | **68.3** |
> | **64** | | | | | | | | | |
> | SelEx | 73.6 | 76.5 | 72.1 | 53.2 | 72.9 | 43.7 | 78.8 | 84.4 | 67.7 |
> | SelEx + Ours | **74.9** | **77.1** | **73.8** | **53.8** | **75.8** | 43.1 | **79.7** | **84.6** | **69.7** |
> | **128** | | | | | | | | | |
> | SelEx | 78.7 | 81.3 | 77.5 | 55.9 | 76.9 | 45.8 | 80.0 | 84.8 | 70.4 |
> | SelEx + Ours | **80.6** | 81.0 | **80.4** | **57.0** | **77.3** | **47.2** | **80.7** | 84.3 | **72.1** |
>
> Our results show a clear trend that as batch size decreases, the performance of baseline methods drops substantially. This is expected because smaller batches produce a less stable and less informative learning signal for both contrastive and prototype-based approaches. Importantly, **BIA improves performance consistently across all tested batch sizes.** For example, With SimGCD on CUB, reducing the batch size to 32 lowers the baseline *All* accuracy to 49.7%. In this difficult setting, BIA adds **3.4%** points and raises the accuracy to 53.1%. A similar effect appears with SelEx on S-cars, where a batch size of 64 yields a baseline of 53.2%, and BIA increases it to 53.8%. This pattern is consistent across datasets and methods, showing that BIA’s geometric regularization does not rely on large batches. It provides a stable structural prior that helps training remain reliable even when the batch-level Gram matrix is estimated from fewer and noisier samples.

---

> ### Author Response · Authors · 2025-11-22
> **Responses to Reviewer fKEF (Part 8)**
>
> > Q3: Gains on Herbarium19.
>
> When the ground-truth number of classes K is given, Table 2 shows that our method obtains consistent gains on Herbarium19, but when K is not given, the performance of our method on Herbarium19 shown in Table 3 stem from a combination of the dataset's intrinsic challenges and our strict experimental protocol designed to demonstrate generalizability.
>
> **1. Intrinsic difficulty of the Herbarium19 dataset**: Herbarium19 presents a uniquely challenging scenario. As we discuss in Appendix C.3, the dataset suffers from a significant domain gap relative to the ImageNet data used for DINO pre-training. This results in initial feature embeddings of relatively low quality, where the representations of different classes already exhibit substantial overlap. BIA functions as a geometric restorer, designed to enhance the capacity of an already meaningful feature space. However, when the foundational embeddings lack sufficient separability to begin with, the ability of any geometric regularizer to construct clean, well-defined manifolds is inherently limited. In this challenging regime, BIA provides a comparable performance while importantly not harming the baseline performance.
>
> **2. A deliberate choice for a unified hyperparameter setting**: A second key factor is our experimental design. To underscore the robust, plug-and-play nature of BIA, we intentionally use **a single, fixed value for the loss weight λ across all datasets and all baseline methods**. We believe this protocol provides a more honest assessment of the method's general applicability, as it avoids dataset-specific hyperparameter tuning or "cherry-picking" results. We note that even on Herbarium19, a mild re-tuning of λ (for instance, reducing it by half) does yield consistent accuracy gains of over 0.5% across the All, Old, and New categories. We present the results from the unified setting in the paper to transparently demonstrate BIA's strong out-of-the-box performance without requiring specialized tuning for each individual benchmark.
>
> ---
> We thank the reviewer for the valuable and constructive comments and suggestions. We are eager to address any additional questions or thoughts the reviewer might have and hope to foster continued communication throughout this process.

---

### Author Response · Authors · 2025-11-22
**General Responses to All Reviewers**

Dear reviewers,

We thank the reviewers for their very helpful and constructive reviews. We appreciate the time and effort the reviewers have dedicated to providing valuable feedback on our work. In response to their comments, we have conducted extensive new experiments and analyses to clarify our contributions, validate our method's robustness, and strengthen the manuscript (revised content is displayed in blue font).

We follow up with responses to each reviewer's individual questions and concerns. Below is a summary of the **main** points addressed for each reviewer. The reviewers are welcome to discuss any lingering questions they may have with us at any time.


**Reviewer fkEF**
*   Revisions to motivational figures (Fig 1 & 2) for improved clarity on the problem and our method's workflow.
*   An in-depth analysis of other isotropy regularizers, including extensive hyperparameter tuning for VICReg and new experiments with direct whitening objectives.
*   Strengthened connections between our initial hypothesis and the quantitative evidence in the paper.
*   New ablation studies on the impact of batch size, demonstrating BIA's effectiveness in challenging regimes.

**Reviewer SYXh**
*   Theoretical justification for the equivalence between the Bures distance and the nuclear norm objective under standard normalization.
*   Comprehensive new robustness tests under varying batch sizes, significant label noise, and class imbalance.
*   New ablation studies incorporating patch tokens to demonstrate BIA's general applicability.
*   Theoretical and empirical comparison with direct whitening objectives (Iso-Frob, Iso-Ent).

**Reviewer aYVi**
*   Clarification on the distinct roles of BIA as a local optimization objective and von Neumann Entropy as a global diagnostic metric.
*   A more detailed comparison with SSL methods, supported by a new, comprehensive hyperparameter analysis for VICReg.
*   Explanation of the BIA loss function's differentiability and its practical implementation.

**Reviewer trb7**
*   Clarification on the novelty and key distinctions of BIA compared to self-supervised anti-collapse methods.
*   Detailed analysis of how BIA dynamically corrects the feature geometry during training.
*   Quantitative evaluation of BIA's low computational overhead.

Cordially,

Authors of BIA

---

### Author Response · Authors · 2025-12-02
**Summary of Post-Rebuttal Score Updates and Context**

We sincerely thank the reviewers and area chairs for their valuable time and thoughtful feedback.

The initial review scores were **(6, 8, 4, 6)**. Following our rebuttal, Reviewer **SYXh** stated in their post-rebuttal comment that our response largely addresses their concerns, and raised their score from 4 to 6 on November 23 to support acceptance. Also, Reviewer **aYVi** noted in their November 23 follow-up that their concerns were addressed and confirmed their positive high rating score (8).

Unfortunately, these updated scores **(6, 8, 6, 6)** are not reflected in the current system due to global score reversion resulting from the early closure of the discussion phase. However, the reviewers’ post-rebuttal comments are still available for review. We provide this clarification to ensure full transparency and thank you for your time and consideration.

---

### Meta-Review · Area_Chair_CSKL · 2025-12-27

**Summary:**

The paper introduces Bures–Isotropy Alignment for generalized category discovery (GCD) and measures the Bures distance from the batch class-token Gram matrix to the identity. The proposed BIA improves accuracy and class-count estimation across multiple baselines and datasets. All the reviewers tend to accept this manuscript, therefore I recommend accepting this paper.

**Reviewer Concerns:**

All the concerns are addressed well by the authors’ rebuttal.

**Reviewer Scores:**

The initial review scores were (6, 8, 4, 6). Following the rebuttal, Reviewer SYXh decided to raise their score from 4 to 6.

---

### Decision · Program_Chairs · 2026-01-26

Accept (Poster)